# BOOSTING THE CYCLE COUNTING POWER OF GRAPH NEURAL NETWORKS WITH $I^2$-GNNS

**Yinan Huang**[1], **Xingang Peng**[1,2], **Jianzhu Ma**[3], **Muhan Zhang**[1]
[1]Institute for Artificial Intelligence, Peking University
[2]School of Intelligence Science and Techology, Peking University
[3]Institute for AI Industry Research, Tsinghua University
`{yinan8114,xingang.peng}@gmail.com`
`majianzhu@tsinghua.edu.cn, muhan@pku.edu.cn`

## ABSTRACT

Message Passing Neural Networks (MPNNs) are a widely used class of Graph Neural Networks (GNNs). The limited representational power of MPNNs inspires the study of provably powerful GNN architectures. However, knowing one model is more powerful than another gives little insight about what functions they can or cannot express. It is still unclear whether these models are able to approximate specific functions such as counting certain graph substructures, which is essential for applications in biology, chemistry and social network analysis. Motivated by this, we propose to study the counting power of Subgraph MPNNs, a recent and popular class of powerful GNN models that extract rooted subgraphs for each node, assign the root node a unique identifier and encode the root node's representation within its rooted subgraph. Specifically, we prove that Subgraph MPNNs fail to count more-than-4-cycles at node level, implying that node representations cannot correctly encode the surrounding substructures like ring systems with more than four atoms. To overcome this limitation, we propose $I^2$-GNNs to extend Subgraph MPNNs by assigning different identifiers for the root node and its neighbors in each subgraph. $I^2$-GNNs' discriminative power is shown to be strictly stronger than Subgraph MPNNs and partially stronger than the 3-WL test. More importantly, $I^2$-GNNs are proven capable of counting all 3, 4, 5 and 6-cycles, covering common substructures like benzene rings in organic chemistry, while still keeping linear complexity. To the best of our knowledge, it is the first linear-time GNN model that can count 6-cycles with theoretical guarantees. We validate its counting power in cycle counting tasks and demonstrate its competitive performance in molecular prediction benchmarks.

## 1 INTRODUCTION

Relational and structured data are usually represented by graphs. Representation learning over graphs with Graph Neural Networks (GNNs) has achieved remarkable results in drug discovery, computational chemistry, combinatorial optimization and social network analysis (Bronstein et al., 2017; Duvenaud et al., 2015; Khalil et al., 2017; Kipf & Welling, 2016; Stokes et al., 2020; You et al., 2018; Zhang & Chen, 2018). Among various GNNs, Message Passing Neural Network (MPNN) is one of the most commonly used GNNs (Zhou et al., 2020; Veličković et al., 2017; Scarselli et al., 2008). However, the representational power of MPNNs is shown to be limited by the Weisfeiler-Lehman (WL) test (Xu et al., 2018; Morris et al., 2019), a classical algorithm for graph isomorphism test. MPNNs cannot recognize even some simple substructures like cycles (Chen et al., 2020). It leads to increasing attention on studying the representational power of different GNNs and designing more powerful GNN models.

The representational power of a GNN model can be evaluated from two perspectives. One is the ability to distinguish a pair of non-isomorphic graphs, i.e., discriminative power. Chen et al. (2019) show the equivalence between distinguishing all pairs of non-isomorphic graphs and approximating all permutation invariant functions (universal approximation). Though the discriminative power provides a way to compare different models, for most GNN models without universal approximation

property, it fails to tell what functions these models can or cannot express. Another perspective is to characterize the function classes expressed by a GNN model. In this regard, Chen et al. (2020) discusses the WL test's power of counting general graph substructures. Graph substructures are important as they are closely related to tasks in chemistry (Deshpande et al., 2002; Jin et al., 2018; Murray & Rees, 2009), biology (Koyutürk et al., 2004) and social network analysis (Jiang et al., 2010). Particularly, cycles play an essential role in organic chemistry. Different types of rings impact the compounds' stability, aromaticity and other chemical properties. Therefore, studying the approximating power of counting substructures, especially cycles, provides a fine-grained and intuitive description of models' representational power and gives insight to real-world practices.

Nevertheless, the difficulty of counting cycles is usually underestimated. Although You et al. (2021) claim that ID-GNNs can count arbitrary cycles at node level, the proof turns out to be incorrect, since it confuses walks with paths (a cycle is a closed path without repeated nodes while walks allow repeated nodes). In fact, even powerful 2-FWL test with cubic complexity can only count up to 7-cycles (Fürer, 2017; Arvind et al., 2020). The difficulty makes us question whether existing powerful models, such as ID-GNNs, can count cycles properly.

ID-GNNs can be categorized into a new class of GNNs named Subgraph GNNs (Cotta et al., 2021; Bevilacqua et al., 2021; Zhang & Li, 2021; You et al., 2021; Zhao et al., 2021; Papp et al., 2021). The core idea is to decompose a graph into a bag of subgraphs and encode the graph by aggregating subgraph representations, though the strategy of extracting subgraphs varies. See Frasca et al. (2022); Papp & Wattenhofer (2022) for detailed discussions. Subgraph GNNs have demonstrated their impressive performance by achieving state-of-the-art results on multiple open benchmarks. Theoretically, the discriminative power of existing Subgraph GNNs is known to be strictly stronger than WL test and weaker than 3-WL test (Frasca et al., 2022). However, it is fair to say we still do not know the approximation power of Subgraph GNNs in terms of counting substructures.

**Main contributions**. In our work, we propose to study the representational power of Subgraph GNNs via the ability to count a specific class of substructures—cycles and paths, because they are the bases to represent some important substructures such as ring systems in chemistry. We focus on Subgraph MPNNs, a subclass of Subgraph GNNs covering Cotta et al. (2021); Zhang & Li (2021); You et al. (2021). Our main contribution include

- We prove that Subgraph MPNNs can count 3-cycles and 4-cycles, but cannot count 5-cycle or any longer cycles at node level. This result is unsatisfying because only a small portion of ring systems are 4-cycles. It also negates the previous proposition that ID-GNNs can use node representations to count arbitrary cycles (You et al., 2021).

- To overcome the limitation, we propose $I^2$-GNNs that extend Subgraph MPNNs by using multiple node identifiers. The main idea is to tie each subgraph with a node pair, including a root node and one of its neighbors. For each resulting subgraph we label the node pair with unique identifiers, which is the key to increasing the representational power.

- Theoretically, we prove that $I^2$-GNNs are strictly more powerful than WL test and Subgraph MPNNs, and partially more powerful than 3-WL test. Importantly, we prove $I^2$-GNNs can count all cycles with length less than 7, covering important ring systems like benzene rings in chemistry. Given bounded node degree, $I^2$-GNNs have linear space and time complexity w.r.t. the number of nodes, making it very scalable in real-world applications. To our best knowledge, $I^2$-GNN is the first linear-time GNN model that can count 6-cycles with rigorous theoretical guarantees. Finally, we validate the counting power of $I^2$-GNNs on both synthetic and real-world datasets. We demonstrate the highly competitive results of $I^2$-GNNs on multiple open benchmarks compared to other state-of-the-art models.

## 2 PRELIMINARIES

Let $G = (V, E)$ be a simple and undirected graph where $V = \{1, 2, 3, ..., N\}$ is the node set and $E \subseteq V \times V$ is the edge set. We use $x_i$ to denote attributes of node $i$ and $e_{i,j}$ to denote attributes of edge $(i, j)$. We denote the neighbors of node $i$ by $N(i) \triangleq \{j \in V | (i, j) \in E\}$. A subgraph $G_S = (V_S, E_S)$ of $G$ is a graph with $V_S \subseteq V$ and $E_S \subseteq E$.

In this paper, we focus on counting paths and cycles. A (simple) $L$-path is a sequence of edges $[(i_1, i_2), (i_2, i_3), ..., (i_L, i_{L+1})]$ such that all nodes are distinct: $i_1 \neq i_2 \neq ... \neq i_{L+1}$. A (simple) $L$-cycle is an $L$-path except that $i_1 = i_{L+1}$. Obviously, we have $L \geq 3$ for any cycle. Two paths/cycles

are considered equivalent if their sets of edges are equal. The count of a substructure $S$ ($L$-cycle or $L$-path) of a graph $G$, denoted by $C(S, G)$, is the total number of inequivalent substructures occurred as subgraphs of the graph. The count of substructures $S$ of a node $i$, denoted by $C(S, i, G)$, is the total number of inequivalent substructures involving node $i$. Specifically, we define $C(\text{cycle}, i, G)$ by number of cycles containing node $i$, and $C(\text{path}, i, G)$ by the number of paths starting from $i$.

Below we formally define the counting task at both graph and node level via distinguishing power. The definitions are similar to the approximation of counting functions (Chen et al., 2020).

**Definition 2.1** (Graph-level counting). *Let $\mathcal{G}$ be the set of all graphs and $\mathcal{F}_{graph}$ be a function class over graphs, i.e., $f : \mathcal{G} \to \mathbb{R}$ for $f \in \mathcal{F}_{graph}$. We say $\mathcal{F}_{graph}$ can count a substructure $S$ at graph level, if for all pairs of graphs $G_1, G_2 \in \mathcal{G}$ satisfying $C(S, G_1) \neq C(S, G_2)$, there exists a model $f \in \mathcal{F}_{graph}$ such that $f(G_1) \neq f(G_2)$.*

**Definition 2.2** (Node-level counting). *Let $\mathcal{G}$ be the set of all graphs, $\mathcal{V}$ be the space of nodes, and $\mathcal{F}_{node}$ be a function class over node-graph tuples, i.e., $f : \mathcal{V} \times \mathcal{G} \to \mathbb{R}$ for $f \in \mathcal{F}_{node}$. We say $\mathcal{F}_{node}$ can count a substructure $S$ at node level, if for all pairs of node-graph tuples $(i_1, G_1), (i_2, G_2)$ satisfying $C(S, i_1, G_1) \neq C(S, i_2, G_2)$, there exists a model $f \in \mathcal{F}_{node}$ such that $f(i_1, G_1) \neq f(i_2, G_2)$.*

It is worth noticing that node-level counting requires stronger approximation power than graph-level counting. This is because the number of substructures of a graph is determined by the number of substructures of nodes, e.g., $C(\text{3-cycle}, G) = \frac{1}{3} \sum_{i \in V} C(\text{3-cycle}, i, G)$ and $C(\text{3-path}, G) = \sum_{i \in V} C(\text{3-path}, i, G)$. Therefore, being able to count a substructure at node level implies the same power at graph level, but the opposite is not true.

## 3   COUNTING POWER OF MPNNs AND SUBGRAPH MPNNs

### 3.1   COUNTING POWER OF MPNNs

Message passing Neural Networks (MPNNs) are a class of GNNs that updates node representations by iteratively aggregating information from neighbors (Gilmer et al., 2017). Concretely, let $h_i^{(t)}$ be the representation of node $i$ at iteration $t$. MPNNs update node representations using

$$\forall i \in V, \quad h_i^{(t+1)} = U_t\left(h_i^{(t)}, \sum_{j \in N(i)} m_{i,j}\right), \text{ where } m_{i,j} = M_t\left(h_i^{(t)}, h_j^{(t)}, e_{i,j}\right). \tag{1}$$

Here $U_t$ and $M_t$ are two learnable functions shared across nodes. Node representations are initialized by node attributes: $h_i^{(0)} = x_i$, or 1 when no node attributes are available. After $T$ iterations, the final node representations $h_i \triangleq h_i^{(T)}$ are passed into a readout function to obtain a graph representation:

$$h_G = R_{\text{graph}}\left(\{h_i | i \in V\}\right). \tag{2}$$

It is known that MPNNs' power of counting substructures is poor: MPNNs cannot count any cycles or paths longer than 2.

**Remark 3.1.** *MPNNs cannot count any cycles at graph level (using $h_G$), and cannot count more-than-2-paths at graph level (using $h_G$). This can be easily seen by considering two graphs: $G_1$ is two unconnected $L$-cycles and $G_2$ is a $2L$-cycle. Any MPNN cannot distinguish them, but $G_1$ and $G_2$ have different numbers of $L$-cycles and $L$-paths. See Appendix B for detailed discussions.*

### 3.2   COUNTING POWER OF SUBGRAPH MPNNs

Subgraph GNNs is to factorize a graph into subgraphs by some pre-defined strategies and aggregate subgraph representations into graph representations. Particularly, a node-based strategy means each subgraph $G_i = (V_i, E_i)$ is tied with a corresponding node $i$. The subgraph $G_i$ is usually called the rooted subgraph of root node $i$. The initial node features of node $j$ in rooted subgraph $G_i$ is $x_{i,j} \triangleq x_j \oplus z_{i,j}$, where $\oplus$ denotes concatenation, $x_j$ denotes the raw node attributes and $z_{i,j}$ are some hand-crafted node labeling. We formally define **Subgraph MPNNs**, a concrete implementation of Subgraph GNNs, including several popular node-based strategies.

**Definition 3.1** (Subgraph MPNNs). *Subgraph MPNNs are a subset of Subgraph GNNs which use a combination of node-based strategies listed below:*

- *Subgraph extraction: **node deletion** $(V_i, E_i) = (V \setminus \{i\}, E \setminus \{(i, j) | j \in N(i)\})$ or **K-hop ego-network** $(V_i, E_i) = EGO_K(i, V, E)$, where $EGO_K(i, V, E)$ is the subgraph induced by nodes within $k$ hops from $i$;*
- *Node labeling: **identity labeling** $z_{i,j} = \mathbb{1}_{i=j}$ or **shortest path distance** $z_{i,j} = spd(i, j)$;*

*and use MPNNs as base GNNs to encode subgraph representations (see Equation (3)). Here $\mathbb{1}$ is used to denote indicator function.*

Concretely, let $h_{i,j}^{(t)}$ be the representation of node $j$ in subgraph $i$ at iteration $t$. Subgraph MPNNs follow a message passing scheme on each subgraph:

$$\forall i \in V, \forall j \in V_i, \quad h_{i,j}^{(t+1)} = U_t \left( h_{i,j}^{(t)}, \sum_{k \in N_i(j)} m_{i,j,k} \right), \text{ where } m_{i,j,k} = M_t \left( h_{i,j}^{(t)}, h_{i,k}^{(t)}.e_{j,k} \right),$$
(3)

Here $N_i(j) \triangleq \{k \in V_i | (k, j) \in E_i\}$ represents node $j$'s neighbors in subgraph $G_i$, and $U_t, M_t$ are shared learnable functions. After $T$ iterations, the final representations $h_{i,j} \triangleq h_{i,j}^{(T)}$ will be passed to a node readout function to obtain node representations $h_i$:

$$\forall i \in V, \quad h_i = R_{\text{node}}(\{h_{i,j} | j \in V_i\}).$$
(4)

Then $\{h_i\}_i$ are further passed to a graph readout function to obtain the graph representation $h_G$:

$$h_G = R_{\text{graph}}(\{h_i | i \in V\}).$$
(5)

Note that Subgraph MPNNs defined in Definition 3.1 and Equations (3), (4), (5) cover previous methods such as $(N - 1)$-Reconstruction GNNs (Cotta et al., 2021), ID-GNNs (You et al., 2021), DS-GNNs (Bevilacqua et al., 2021) and Nested GNNs (Zhang & Li, 2021), though there exists other variants of Subgraph GNNs (Zhao et al., 2021; Bevilacqua et al., 2021; Frasca et al., 2022; Qian et al., 2022) that do not fit into Subgraph MPNNs. Subgraph MPNNs are strictly more powerful than MPNNs, because (1) Subgraph MPNNs reduce to $T$-layer MPNNs by taking $T$-hop ego-network without any node labeling, performing $T$-layers of message passing and using root node pooling $h_i = h_{i,i}$; and (2) Subgraph MPNNs can distinguish pairs of regular graphs which MPNNs cannot distinguish (e.g., graphs in Figure 2).

Although Subgraph MPNNs are more expressive than MPNNs, it is still unclear if Subgraph MPNNs can count cycles and paths. Note that the proof in You et al. (2021) confuses paths with walks; see Appendix C. Here we give an upper bound for Subgraph MPNNs' counting power at graph level.

**Proposition 3.1.** *Subgraph MPNNs cannot count 8-cycles and 8-paths at graph level. This can be seen by a pair of strongly regular graphs, the 4x4 Rook's graph and the Shrikhande graph. These two graphs have different numbers of 8-cycles and 8-paths (Fürer, 2017; Arvind et al., 2020), but Subgraph MPNNs fail to distinguish them (see Appendix D.1). The counter-example also supports the conclusion that Subgraph MPNNs are weaker than 3-WL test (Frasca et al., 2022).*

Further, we give a **complete characterization** of Subgraph MPNNs' cycle counting power at node level, shown in the following theorems.

**Theorem 3.1.** *Subgraph MPNNs can count 3-cycles, 4-cycles, 2-paths and 3-paths at node level.*

**Theorem 3.2.** *Subgraph MPNNs cannot count more-than-4-cycles and more-than-3-paths at node level.*

We include the proofs in Appendix D. Theorems 3.1 and 3.2 indicate that although Subgraph MPNNs bring improvement over MPNNs, they still can only count up to 4-cycles. This is far from our expectation to count important substructures like benzene rings.

# 4 $I^2$-GNNs

The limitation of Subgraph MPNNs motivates us to design a model with stronger counting power. A key observation is that assigning the root node a unique identifier can already express all node-based strategies listed in Definition 3.1. See Lemma D.1. Therefore, we conjecture that assigning

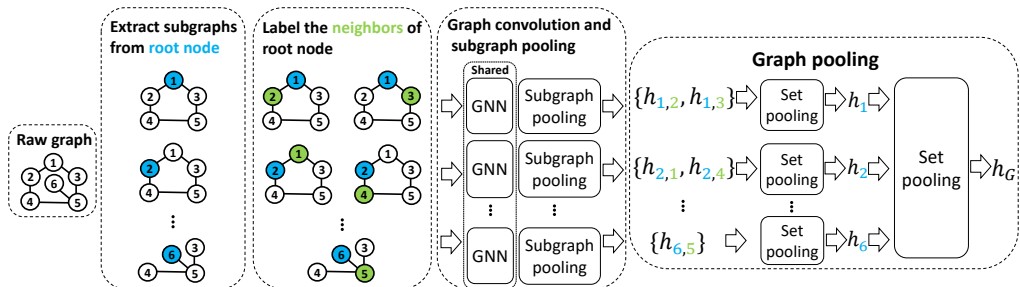

Figure 1: Architecture of I$^2$-GNN. It extracts a $K$-hop ego-network for each root node and iteratively assigns unique identifiers to both the root node (blue) and each of its neighbors (green).

more than one node identifier simultaneously can further lift the representational power. However, naively labeling all pairs of nodes may suffer from poor scalability due to the square complexity, like other high-order GNNs. Fortunately, we notice that substructures like cycles and paths are highly localized. For instance, a 6-cycle must exist in the 3-hop ego-network of the root node. This implies that a local labeling method might be sufficient to boost the counting power.

Given these observations, we adopt a localized identity labeling strategy and propose **I$^2$-GNNs**: except for the unique identifier of root node $i$, we further assign another unique identifier to one of the **neighbors** of the root node, called *branching node $j$*. To preserve the permutation equivariance, the branching node should iterate over all neighbors of the root node. The resulting model essentially associates each connected 2-tuple $(i, j)$ with a subgraph $G_{i,j}$, in which $i, j$ are labeled with the corresponding identifiers. Compared to Subgraph MPNNs, it only increases the complexity by a factor of node degree. See Figure 1 for an illustration of the graph encoding process of I$^2$-GNNs.

Formally, for each root node $i \in V$, we first extract its $K$-hop ego-network with identity labeling. Then for **every** branching node $j \in N(i)$, we further copy the subgraph and assign an additional identifier for $j$. This results in a subgraph $G_{i,j} = (V_i, E_i)$ tied with $(i, j)$, where node attributes are augmented with **two identifiers**: $x_{i,j,k} \triangleq x_k \oplus \mathbb{1}_{k=i} \oplus \mathbb{1}_{k=j}$. Let $h_{i,j,k}^{(t)}$ be the representation of node $k$ in subgraph $G_{i,j}$ at iteration $t$. I$^2$-GNNs use MPNNs in each subgraph:

$$\forall i \in V, \forall j \in N(i), \forall k \in V_i,$$

$$h_{i,j,k}^{(t+1)} = U_t \left( h_{i,j,k}^{(t)}, \sum_{l \in N_i(k)} m_{i,j,k,l} \right), \text{ where } m_{i,j,k,l} = M_t \left( h_{i,j,k}^{(t)}, h_{i,j,l}^{(t)}, e_{k,l} \right). \tag{6}$$

After $T$ iterations we obtain final representations $h_{i,j,k} \triangleq h_{i,j,k}^{(T)}$. We use an edge readout function:

$$\forall i \in V, \forall j \in N(i), \quad h_{i,j} = R_{\text{edge}} \left( \{ h_{i,j,k} | k \in V_i \} \right), \tag{7}$$

a node readout function:

$$\forall i \in V, \quad h_i = R_{\text{node}} \left( \{ h_{i,j} | j \in N(i) \} \right), \tag{8}$$

and finally a graph readout function:

$$h_G = R_{\text{graph}} \left( \{ h_i | i \in V \} \right). \tag{9}$$

Intuitively, I$^2$-GNNs improve the representational power by **breaking the symmetry of the neighbors of root nodes**. In the rooted subgraph $G_i$, the root node $i$ treats the messages from distinct branching nodes $j$ differently. The following discussion demonstrates that I$^2$-GNNs do have stronger discriminative power over MPNNs and Subgraph MPNNs.

**Proposition 4.1.** *I$^2$-GNNs are **strictly more powerful** than Subgraph MPNNs. This can be seen via: (1) I$^2$-GNNs can reduce to Subgraph MPNNs by ignoring the branching node identifier. (2) I$^2$-GNNs can distinguish the 4x4 Rook's graph and the Shrikhande graph, while Subgraph MPNNs and 3-WL test cannot. This also suggests that I$^2$-GNNs are **partially stronger** than the 3-WL test. See Appendix E.1 for more details. Moreover, I$^2$-GNNs' discriminative power is bounded by the 4-WL test as they can be implemented by 4-IGNs, which is analogous to the relation between Subgraph MPNNs and 3-IGNs (Frasca et al., 2022).*

One may ask if the additional identifier of $I^2$-GNNs brings stronger counting power. The answer is yes. We note that in $I^2$-GNNs, the root node $i$ is aware of the first edge $(i, j)$ in each path it is counting, i.e., $(i \rightarrow j \rightarrow ...)$, while for Subgraph MPNNs the first edge in paths is always anonymous, i.e., $(i \rightarrow ...)$. As the first edge is determined, the counting of $L$-paths from $i$ is transformed into an easier one—the counting of $(L-1)$-paths from $j$. Since cycles are essentially a special kind of paths, the cycle counting power is boosted too. However, naively applying this argument only shows that $I^2$-GNNs can count up to 5-cycles (one more than Subgraph MPNNs). Our main theorem below indicates that $I^2$-GNNs' cycle counting power is actually **boosted to at least** 6**-cycles**.

**Theorem 4.1.** *$I^2$-GNNs can count* 3*,* 4*,* 5 *and* 6*-cycles at node level.*

We include all proofs in Appendix E. Theorem 4.1 provides a lower bound of $I^2$-GNNs' cycle counting power. Recall that the cycle counting power of the 3-WL test is upper bounded by 7-cycles, while the linear-time $I^2$-GNNs can already approach that. The significance of Theorem 4.1 is that it indicates the feasibility of using a **local and scalable** model to encode cycle substructures, rather than applying global algorithms such as $k$-WL test or relational pooling with exponential cost.

Below we also show some positive results of counting other substructures with $I^2$-GNNs.

**Theorem 4.2.** *$I^2$-GNNs can count* 3 *and* 4*-paths at node level.*

**Theorem 4.3.** *$I^2$-GNNs can count all connected graphlets with size 3 or 4 at node level.*

See the definition of node-level graphlets counting in Appendix Figure 6. Note that connected graphlets with size 4 includes 4-cliques, a substructure that even 3-WL test cannot count (Fürer, 2017). This again verifies $I^2$-GNNs' partially stronger power than 3-WL.

## 5 RELATED WORKS

**WL-based GNNs.** Xu et al. (2018) and Morris et al. (2019) showed that MPNNs are at most as powerful as the WL test in terms of distinguishing non-isomorphic graphs, which motivates researchers to propose provably more powerful GNN models. One line of research is to extend MPNNs to simulate high-dimensional WL test by performing computations on $k$-tuples. On one hand, Morris et al. (2019) proposes $k$-dimensional GNNs called $k$-GNNs, which apply message passing between $k$-tuples and can be seen as a local and neural variant of $k$-WL test. Morris et al. (2020b) further extend $k$-GNNs by aggregating only from local neighbors, obtaining a scalable and strictly more powerful model. Despite the good scalability due to the localized nature, it is still unclear if these models can achieve the discriminative power of the 3-WL test. On the other hand, Maron et al. (2019a) proposed a tensor-based model that is provably as powerful as the 3-WL test. The downside is its cubic complexity.

**Permutational invariant and equivariant GNNs.** There is a line of research studying GNNs from the perspective of equivariance to permutation group. In this regard, Maron et al. (2018) proposed Invariant Graph Networks (IGNs), which apply permutational equivariant linear layers with point-wise nonlinearity to the input adjacency matrix. $k$-IGNs (using at most $k$-order tensors) are shown to have equivalent discriminative power to $k$-WL test (Maron et al., 2019a; Geerts, 2020; Azizian & Lelarge, 2020). With sufficiently large $k$, $k$-IGNs can reach universal approximation to any invariant functions on graphs (Maron et al., 2019b; Azizian & Lelarge, 2020). Relational pooling (Murphy et al., 2019) utilizes the Reynold operator, a linear map that projects arbitrary functions to the invariant function space by averaging over the permutation group. Relational pooling is shown to be a universal approximator for invariant functions, but it does not scale as the size of the permutation group grows factorially with graph size.

**Subgraph GNNs.** Another line of research aims at breaking the limit of MPNNs by encoding node representations from subgraphs rather than subtrees. In this respect, Abu-El-Haija et al. (2019); Tahmasebi et al. (2020); Sandfelder et al. (2021); Nikolentzos et al. (2020) study convolutions on $K$-hop neighbors instead of only 1-hop neighbors. See Feng et al. (2022) for an analysis of their power. Wijesinghe & Wang (2021) weights the message passing based on the subgraph overlap. On the other hand, a recent and popular class of GNNs dubbed Subgraph GNNs views a graph as a bag of subgraphs. Subgraph extraction policies vary among these works, with possible options including node and/or edge deletion (Cotta et al., 2021; Bevilacqua et al., 2021; Papp et al., 2021), node identity augmentation (You et al., 2021) and ego-network extraction (Zhang & Li, 2021; Zhao

et al., 2021). The base GNNs are also flexible, varying from MPNNs to relational pooling (Chen et al., 2020). A contemporary work (Frasca et al., 2022) further explores the theoretical upper bound of the representational power of Subgraph MPNNs, showing that all existing Subgraph MPNNs can be implemented by 3-IGNs and thus are weaker than the 3-WL test. Labeling trick (Zhang et al., 2021; Zhang & Chen, 2018) uses multi-node labeling for muti-node task, but it cannot directly generate equivariant single-node/graph representation.

**Feature-augmented GNNs.** Some works improve expressiveness by augmenting node features. Bouritsas et al. (2022); Barceló et al. (2021) augment the initial node features with hand-crafted structural information encoded in the surrounding subgraphs, and Dwivedi et al. (2021) adopts positional encoding and devises message passing for positional embedding. Loukas (2019; 2020) instead study the representational power in case where all nodes have uniquely identified features.

**GNNs' power of counting graph substructures.** The ability to count graph substructures is another perspective of studying GNNs' representational power. Many previous works characterized the power to count substructures for the WL test and variants of GNNs. Fürer (2017); Arvind et al. (2020) give a complete description of 1-WL combinatorial invariants (i.e., all substructures that can be counted by the 1-WL test) and a partial result for 2-FWL. Particularly, the power of counting cycles and paths of the 2-FWL test is fully understood: the 2-FWL test can and only can count up to 7-cycles and 7-paths. Chen et al. (2020) study the counting power of induced subgraph counting and give general results of $k$-WL test, but the bound is loose. Tahmasebi et al. (2020) study the counting power of Recursive Neighborhood Pooling GNNs and give the complexity lower bound of counting substructures for generic algorithms.

# 6 EXPERIMENTS

This section aims to validate our theoretical results and study $I^2$-GNNs' empirical performance (https://github.com/GraphPKU/I2GNN.). Particularly, we focus on the following questions:
**Q1**: Does the discriminative power of $I^2$-GNNs increase compared to Subgraph MPNNs?
**Q2**: Can $I^2$-GNNs reach their theoretical counting power?
**Q3**: How do $I^2$-GNNs perform compared to MPNNs, Subgraph MPNNs and other state-of-the-art GNN models on open benchmarks for graphs?

Note that in the experiments, shortest path distance (SPD) labeling (Li et al., 2020) is uniformly used as an alternative to identity labeling in $I^2$-GNNs. SPD labeling is also used in Nested GNNs (Zhang & Li, 2021) and GNNAK (Zhao et al., 2021). Theoretically SPD labeling has the same representational power as identity labeling. We study the effects of SPD labeling in Appendix 10.

## 6.1 DISCRIMINATING NON-ISOMORPHIC GRAPHS

**Datasets.** To answer Q1, we study the discriminative power on two synthetic datasets: (1) EXP (Abboud et al., 2020), containing 600 pairs of non-isomorphic graphs that cannot be distinguished by the 1-WL/2-WL test; (2) SR25 (Balcilar et al., 2021), containing 150 pairs of non-isomorphic strongly regular graphs that 3-WL fails to distinguish. We follow the evaluation process in (Balcilar et al., 2021) that compares the graph representations and reports successful distinguishing cases.

Table 1: Accuracy on EXP/SR25.

| Method | EXP | SR25 |
|--------|------|------|
| Base GNN | 0% | 0% |
| ID-GNN | 100% | 0% |
| NGNN | 100% | 0% |
| GNNAK+ | 100% | 0% |
| PPGN | 100% | 0% |
| $I^2$-GNN | 100% | **100%** |

**Models.** Adopting GIN (Xu et al., 2018) as the base GNN, we compare $I^2$-GNNs to ID-GNNs (You et al., 2021), Nested GNNs (NGNNs) (Zhang & Li, 2021) and GNNAK+ (Zhao et al., 2021). These Subgraph GNNs are known to be strictly stronger than 1-WL but weaker than 3-WL. We also consider PPGN (Maron et al., 2019a) known to be as powerful as 3-WL.
**Results.** Table 1 shows that all models except $I^2$-GNN fail the SR25 dataset with 0% accuracy. In contrast, $I^2$-GNN achieves a 100% accuracy. It supports Proposition 4.1 that $I^2$-GNNs is strictly stronger than Subgraph MPNNs and partially stronger than the 3-WL test.

Table 2: Normalized MAE results of counting cycles at node level on synthetic and ChEMBL dataset. The colored cell means an error less than $0.01$ (synthetic) or $0.001$ (ChEMBL).

| Method | Synthetic (norm. MAE) | | | | ChEMBL (norm. MAE) | | | |
|---|---|---|---|---|---|---|---|---|
| | 3-Cycle | 4-Cycle | 5-Cycle | 6-Cycle | 3-Cycle | 4-Cycle | 5-Cycle | 6-Cycle |
| Base GNN | 0.3515 | 0.2742 | 0.2088 | 0.1555 | 0.1326 | 0.0780 | 0.4307 | 0.4268 |
| ID-GNN | 0.0006 | 0.0022 | 0.0490 | 0.0495 | 0.0001 | 0.0008 | 0.0006 | 0.0024 |
| NGNN | 0.0003 | 0.0013 | 0.0402 | 0.0439 | 0.0001 | 0.0005 | 0.0003 | 0.0053 |
| GNNAK+ | 0.0004 | 0.0041 | 0.0133 | 0.0238 | 0.0001 | 0.0011 | 0.0002 | 0.0006 |
| PPGN | 0.0003 | 0.0009 | 0.0036 | 0.0071 | 0.0001 | 0.0169 | 0.0001 | 0.0007 |
| $I^2$-GNN | 0.0003 | 0.0016 | 0.0028 | 0.0082 | 0.0001 | 0.0005 | 0.0001 | 0.0003 |

Table 3: Normalized MAE results of counting graphlets at node level on synthetic dataset.

| Method | Synthetic (norm. MAE) | | | | |
|---|---|---|---|---|---|
| | Tailed Triangle | Chordal Cycle | 4-Clique | 4-Path | Triangle-Rectangle |
| Base GNN | 0.3631 | 0.3114 | 0.1645 | 0.1592 | 0.2979 |
| ID-GNN | 0.1053 | 0.0454 | 0.0026 | 0.0273 | 0.0628 |
| NGNN | 0.1044 | 0.0392 | 0.0045 | 0.0244 | 0.0729 |
| GNNAK+ | 0.0043 | 0.0112 | 0.0049 | 0.0075 | 0.1311 |
| PPGN | 0.0026 | 0.0015 | 0.1646 | 0.0041 | 0.0144 |
| $I^2$-GNN | **0.0011** | **0.0010** | **0.0003** | **0.0041** | **0.0013** |

## 6.2 GRAPH SUBSTRUCTURE COUNTING

**Datasets.** To answer Q2, we adopt the synthetic dataset from Zhao et al. (2021) and a bioactive molecules dataset named ChEMBL (Gaulton et al., 2012) to perform node-level counting tasks. The synthetic dataset contains 5,000 graphs generated from different distributions. The training/validation/test spliting is 0.3/0.2/0.5. The ChEMBL dataset is filtered to contain 16,200 molecules with low fingerprint similarity. The task is to perform node-level regression on the number of 3-cycles, 4-cycles, 5-cycles, 6-cycles, tailed triangles, chordal cycles, 4-cliques, 4-paths and triangle-rectangles respectively (continuous outputs to approximate discrete labels). See Appendix F.1 for more details about definitions of these graphlets, dataset preprocessing, model implementation and experiment setup.

**Models.** We compare with other Subgraph GNNs, including ID-GNNs, NGNNs and GNNAK+. Besides, PPGNs, which theoretically can count up to 7-cycles, are also included for comparison. We use a 4-layer GatedGNN (Li et al., 2015) as the base GNN to build ID-GNNs, NGNNs and $I^2$-GNNs. GNNAK+ is implemented using 4-layer GNNAK+ layers with 1-layer GatedGCNs as inner base GNNs. PPGNs are realized with 4 blocks of PPGN layers.

**Results.** We run the experiments with three different random seeds and report the average normalized test MAE (i.e. test MAE divided by label standard deviation) in Table 2 and 3. On both datasets, Subgraph MPNNs (NGNNs and ID-GNNs) and $I^2$-GNNs attain a relatively low error ($< 0.01$) for counting of $3, 4$-cycles, which is consistent with Theorems 3.1 and 4.1. On the synthetic dataset, if we compare 5-cycles, 6-cycles to 3-cycles, 4-cycles. The MAE of Subgraph MPNNs get nearly 30 times greater. It supports Theorem 3.2 that Subgraph MPNNs fail to count 5-cycles and 6-cycles at node level. GNNAK+, though not belonging to Subgraph MPNNs, also gets a 3∼6 times greater MAE. In comparison, PPGN and $I^2$-GNNs still keep a stable MAE less than 0.01. On the ChEMBL dataset, the 6-cycle MAE of Subgraph MPNNs also amplifies by 5 times compared to 4-cycles MAE. In contrast, $I^2$-GNNs' MAE almost remains the same. These observations support Theorem 4.1.

**Ablation study.** We study the impact of the key element—the additional identifier, on $I^2$-GNNs' counting power. The results can be found in Appendix 10.

## 6.3 MOLECULAR PROPERTIES PREDICTION

**Datasets.** To answer Q3, we adopt three popular molecular graphs dataset—QM9, ZINC and ogbg-molhiv. QM9 contains 130k small molecules, and the task is regression on twelve targets such as energy. The training/validation/test splitting ratio is 0.8/0.1/0.1. ZINC (Dwivedi et al., 2020), including ZINC-small (10k graphs) and ZINC-large (250k graphs), is a free database of commercially-

Table 4: MAE results on QM9 (smaller the better).

| Target | 1-GNN | 1-2-3-GNN | DTNN | Deep LRP | PPGN | NGNN | $I^2$-GNN |
|--------|-------|-----------|------|----------|------|------|-----------|
| $\mu$ | 0.493 | 0.476 | 0.244 | 0.364 | **0.231** | 0.428 | 0.428 |
| $\alpha$ | 0.78 | 0.27 | 0.95 | 0.298 | 0.382 | 0.29 | **0.230** |
| $\varepsilon_{homo}$ | 0.00321 | 0.00337 | 0.00388 | **0.00254** | 0.00276 | 0.00265 | 0.00261 |
| $\varepsilon_{lumo}$ | 0.00355 | 0.00351 | 0.00512 | 0.00277 | 0.00287 | 0.00297 | **0.00267** |
| $\Delta\varepsilon$ | 0.0049 | 0.0048 | 0.0112 | **0.00353** | 0.00406 | 0.0038 | 0.0038 |
| $R^2$ | 34.1 | 22.9 | 17.0 | 19.3 | **16.07** | 20.5 | 18.64 |
| ZPVE | 0.00124 | 0.00019 | 0.00172 | 0.00055 | 0.0064 | 0.0002 | **0.00014** |
| $U_0$ | 2.32 | **0.0427** | 2.43 | 0.413 | 0.234 | 0.295 | 0.211 |
| $U$ | 2.08 | **0.111** | 2.43 | 0.413 | 0.234 | 0.361 | 0.206 |
| $H$ | 2.23 | **0.0419** | 2.43 | 0.413 | 0.229 | 0.305 | 0.269 |
| $G$ | 1.94 | **0.0469** | 2.43 | 0.413 | 0.238 | 0.489 | 0.261 |
| $C_v$ | 0.27 | 0.0944 | 2.43 | 0.129 | 0.184 | 0.174 | **0.0730** |

Table 5: Four-runs MAE results on ZINC (smaller the better) and ten-runs AUC results on ogbg-molhiv (larger the better). The * indicates the model uses virtual node on ogbg-molhiv.

| Method | ZINC-12K (MAE) | ZINC-250K (MAE) | ogbg-molhiv (AUC) |
|--------|----------------|-----------------|-------------------|
| GIN* | 0.163±0.004 | 0.088±0.002 | 77.07±1.49 |
| PNA | 0.188±0.004 | – | 79.05±1.32 |
| DGN | 0.168±0.003 | – | 79.70±0.97 |
| HIMP | 0.151±0.006 | 0.036±0.002 | 78.80±0.82 |
| GSN | 0.115±0.012 | – | 80.39±0.90 |
| Deep LRP | – | – | 77.19±1.40 |
| CIN-small | 0.094±0.004 | 0.044±0.003 | 80.05±1.04 |
| CIN | **0.079**±0.006 | **0.022**±0.002 | **80.94**±0.57 |
| Nested GIN* | 0.111±0.003 | 0.029±0.001 | 78.34±1.86 |
| GNNAK+ | 0.080±0.001 | – | 79.61±1.19 |
| SUN (EGO) | 0.083±0.003 | – | 80.03±0.55 |
| $I^2$-GNN | 0.083±0.001 | **0.023**±0.001 | 78.68±0.93 |

available chemical compounds, and the task is graph regression. The ogbg-molhiv dataset contains 41k molecules for graph classification. The original release provides the splitting of both ZINC and ogbg-molhiv.

**Models.** For QM9, we adopt baselines including 1-GNN, 1-2-3-GNN (Morris et al., 2019), DTNN (Wu et al., 2018), Deep LRP (Chen et al., 2020), PPGN and NGNN. The baseline results are from Zhang & Li (2021). Note that we omit methods (Klicpera et al., 2020; Anderson et al., 2019; Qiao et al., 2020; Liu et al., 2021) that utilize additional geometric features or quantum mechanics theory, since our goal is to compare the models' graph representation power. We choose NNConv (Gilmer et al., 2017) as our base GNN to construct $I^2$-GNN. For ZINC and ogbg-molhiv, we use GIN as the based GNN. We make comparisons to baselines such as GNNAK+, PNA (Corso et al., 2020), DGN (Beaini et al., 2021), HIMP (Fey et al., 2020), GSN (Bouritsas et al., 2022), Deep LRP (Chen et al., 2020), SUN (Frasca et al., 2022) and CIN (Bodnar et al., 2021). See more details in Appendix F.2.

**Results.** On QM9, Table 4 shows that $I^2$-GNN outperforms NGNN over most targets, with an average 21.6% improvement. Particularly, $I^2$-GNN attains the best results on four targets out of twelve. On ZINC, $I^2$-GNN brings 25% and 18% performance gains to Nested GIN on ZINC-12k and ZINC-250k respectively. Moreover, despite being a model targeting on cycle counting, $I^2$-GNN approaches the SOTA results of CIN. On ogbg-molhiv, $I^2$-GNN improves the AUC of Nested GIN by 0.3%. These results suggest that $I^2$-GNNs also bring improvement to general graph regression/classification tasks.

## 7 CONCLUSION

We propose to study the representational power of Subgraph MPNNs via the ability to count cycles and paths. We prove that Subgraph MPNNs fail to count more-than-4-cycles at node level, which limits their power to encode surrounding ring systems with more than four atoms. Inspired by the localized nature of cycles, we extend Subgraph MPNNs by assigning an additional identifier to the neighbors of the root node to boost the counting power. The resulting model named $I^2$-GNNs turns out to be able to count at least 6-cycles in linear time with theoretical guarantee. Meanwhile, $I^2$-GNNs maintain excellent performance on general graph tasks.

## 8 ACKNOWLEDGE

This project is supported in part by the National Key Research and Development Program of China (No. 2021ZD0114702).

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

## A  WEISFEILER-LEHMAN TEST

The Weisfeiler-Lehman (WL) Test (Weisfeiler & Leman, 1968) is a heuristic algorithm for the graph isomorphism test. Initially, the WL test assigns all nodes the same color, and then it follows an iterative color refinement scheme to update node colors: for each node, the new color is obtained by applying a hash function to its old color and the multiset of its neighbors' old colors. Formally, each iteration can be written as

$$c_i^{(t+1)} = \text{Hash}\left(c_i^{(t)}, \{c_j^{(t)} | j \in N(i)\}\right), \tag{10}$$

where $c_i^{(t)}$ is the color of node $i$ at iteration $t$, and $\{\cdot\}$ should be understood as multiset. The algorithm terminates if the number of colors does not change after one iteration. Two graphs are determined to be non-isomorphic if the histogram of colors differs. The $k$-dimensional Weisfeiler-Lehman ($k$-WL) Test with $k \geq 2$ is a generalized version of WL test that performs color refinement on $k$-tuples. For $k \geq 2$, the discriminative power of the $k$-WL test is shown to increase constantly as $k$ increases (Cai et al., 1992; Maron et al., 2019a). See Morris et al. (2019); Huang & Villar (2021) for more information.

## B  COUNTING POWER OF MPNNS

Table 6: Node-level cycle and path counting power of GNN models.

| Substructures | 2-Path | 3-Path | 4-Path | 3-Cycle | 4-Cycle | 5-Cycle | 6-Cycle |
|---|---|---|---|---|---|---|---|
| MPNNs | ✓ | ✗ | ✗ | ✗ | ✗ | ✗ | ✗ |
| Subgraph MPNNs | ✓ | ✓ | ✗ | ✓ | ✓ | ✗ | ✗ |
| I²-GNNs | ✓ | ✓ | ✓ | ✓ | ✓ | ✓ | ✓ |

In this section we formally prove the counting power of MPNNs. We first give a positive result, that is MPNNs can count 2-paths at node level (thus graph level).

**Theorem B.1.** *MPNNs can count 2-paths at node level.*

*Proof.* To count 2-paths starting from $i$, one can consider the 2-layer message passing function $h_i = \sum_{j \in N(i)} (d_j - 1)$ (degree $d_j$ is learned in the first layer). Then $h_i$ equals to the number of 2-path with $i$ being the endpoint. □

Next we give the negative results at graph level (thus node level).

**Theorem B.2.** *MPNNs cannot count paths with length greater than 2 at graph level, and cannot count any cycles at graph level.*

*Proof.* Let $L$ be any positive integer greater than or equal to 3. Let $G_1$ be two $L$-cycle and $G_2$ be $2L$-cycle. All nodes has a degree of 2, and thus the graph representations are indistinguishable. We see that $G_1$ has two $L$-cycles while $G_2$ does not. Moreover, $G_2$ has $L$-paths while $G_1$ does not. Therefore we finish the proof. □

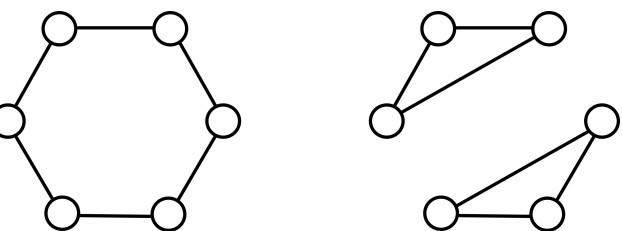

Figure 2: A pair of regular graphs that MPNNs cannot distinguish.

## C DISCUSSION ON PROPOSITION 2 IN YOU ET AL. (2021)

In You et al. (2021), the Proposition 2 claims that an $L$-layer ID-GNN can use node representations to express the number of cycles involving the node. In the proof, they use induction: they assume for arbitrary nodes $i, j$, the representations $h_{i,j}$ equals to the number of $L$-paths from $i$ to $j$, and they try to prove the case of $(L + 1)$-paths using an additional layer of ID-GNN. The basic logic they use is that if there exists an $L$-path from $i$ to $k$, and $k$ is adjacent to $j$, then there must exist an $(L + 1)$-path from $i$ to $j$. However, it is not true because the $L$-path from $i$ to $k$ might contain $j$ already. The induction holds for walks only. In conclusion, the proof actually shows ID-GNNs can count walks, but the number of walks cannot determine the number of cycles.

## D COUNTING POWER OF SUBGRAPH MPNNS

In this section we give formal proofs of Proposition 3.1, Theorem 3.1 and Theorem 3.2. To begin with, we prove some useful lemmas.

**Lemma D.1** (Identity labeling is the most powerful strategy). *Stategy of augmenting root nodes with unique identifiers, i.e. $(V_i, E_i) = (V, E)$ and $x_{i,j} = x_i \oplus \mathbb{1}_{i=j}$, can express all strategies in 3.1.*

*Proof.* For simplicity, we assume $h_{i,j}^{(0)} = \mathbb{1}_{i=j}$. We discuss the listed strategies one by one.

- Node deletion: we can recognize the root node by its identifier and "mask out" the root node $i$ from the subgraph.

- Ego-networks: to mimic a $K$-hop ego-networks extraction, we can apply a $K$-layers MPNNs to mark nodes in the $K$-hop ego-networks. Concretely, $K$ iterations of the following message passing layer label all involving nodes as 1 while others are 0.

$$m_{i,j} = \sum_{k \in N(j)} h_{i,k}^{(t)}, \quad h_{i,j}^{(t+1)} = \mathbb{1}_{h_{i,j}^{(t)}=0} \mathbb{1}_{m_{i,j}>0} + \mathbb{1}_{h_{i,j}^{(t)} \neq 0} \tag{11}$$

  Then we mask out all nodes with zero labels, implementing a $K$-hop ego-networks extraction.

- Shortest path distances: similar to labeling nodes in the $K$-hop ego-netwroks, we can construct the following message passing layer

$$m_{i,j} = \sum_{k \in N(j)} h_{i,k}^{(t)}, \quad h_{i,j}^{(t+1)} = (t+1)\mathbb{1}_{h_{i,j}^{(t)}=0} \mathbb{1}_{m_{i,j}>0} + \mathbb{1}_{h_{i,j}^{(t)} \neq 0} \tag{12}$$

$\square$

In the following discussions, we refer Subgraph MPNNs to this specific model: Subgraph MPNNs with identity node labeling. We assume the identifier of root node is always available in subgraph $G_i$ throughout.

**Lemma D.2.** *Subgraph MPNNs can count 2-paths and 3-paths that starts from node $i$ and ends at node $j$ using $h_{i,j}$.*

*Proof.* We can construct the following MPNNs to count 2-path between the root node $i$ and another node $j$:

$$h_{i,j}^{(1)} = \sum_{k \in N(j))} \mathbb{1}_{k=i}. \tag{13a}$$

$$h_{i,j}^{(2)} = \mathbb{1}_{j \neq i} \cdot \sum_{k \in N(j)} h_{i,k}^{(1)}. \tag{13b}$$

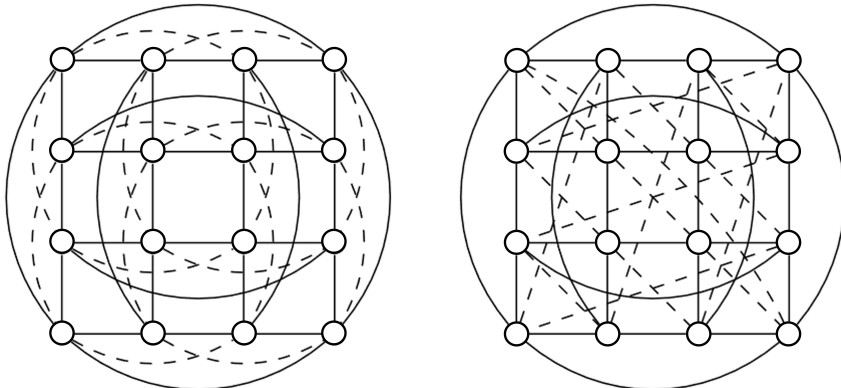

Figure 3: The 4x4 Rook's graph and the Shrikhande graph (some edges are dashed just to ensure readability). The figure is modified from Arvind et al. (2020).

One can easily see that $h_{i,j}^{(2)}$ equals to number of 2-paths from $i$ to $j$. Similarly, we construct the following MPNNs to count 3-paths between node $i$ and $j$:

$$h_{i,j}^{(1)} = \sum_{k \in N_i(j)} \mathbb{1}_{k=i}. \tag{14a}$$

$$h_{i,j}^{(2)} = \mathbb{1}_{j \neq i} \sum_{k \in N(j)} h_{i,k}^{(1)}. \tag{14b}$$

$$h_{i,j}^{(3)} = \mathbb{1}_{j \neq i} \sum_{k \in N(j)} \left( h_{i,k}^{(2)} - h_{i,j}^{(1)} \right). \tag{14c}$$

To see why this is true, note that $h_{i,j}^{(1)}$ marks the neighbors of $i$ and $h_{i,j}^{(2)}$ marks the 2-hop neighbors of $i$. One one hand, if node $j$ is not the neighbor of $i$, then $\sum_{k \in N(j)} h_{i,j}^{(2)}$ is exactly the number of 3-paths from $i$ to $j$. On the other hand, if $j$ is the neighbor of $i$, then $\sum_{k \in N(j)} h_{i,k}^{(2)}$ will additionally get $d_j - 1$ times unexpected counts. Thus by subtraction, we can see $\sum_{k \in N(j)} h_{i,j}^{(2)} - \mathbb{1}_{j \in N(i)} (d_j - 1) = \sum_{k \in N(j)} (h_{i,j}^{(2)} - h_{i,j}^{(1)})$ correctly counts 3-paths from $i$ to $j$. $\qquad \square$

### D.1 PROOF OF PROPOSITION 3.1

Consider the 4x4 Rook's graph and the Shrikhande graph shown in Figure 3. We are going to show that Subgraph MPNNs cannot distinguish them. We choose the node in top left coner as the root node. We use a matrix to denote the colors of nodes in this 4x4 grid. Initially, in both graphs let the root node's color be 1, and other nodes are 0:

$$h_{\text{Rook}}^{(0)} = h_{\text{Shrik}}^{(0)} = \begin{pmatrix} 1 & 0 & 0 & 0 \\ 0 & 0 & 0 & 0 \\ 0 & 0 & 0 & 0 \\ 0 & 0 & 0 & 0 \end{pmatrix}. \tag{15}$$

In the first iteration, let the hash function be $\text{Hash}(1, \{0,0,0,0,0,0\}) = 2$,, $\text{Hash}(0, \{1,0,0,0,0,0\}) = 1$ and $\text{Hash}(0, \{0,0,0,0,0,0\}) = 0$. Then the new colors become:

$$h_{\text{Rook}}^{(1)} = \begin{pmatrix} 2 & 1 & 1 & 1 \\ 1 & 0 & 0 & 0 \\ 1 & 0 & 0 & 0 \\ 1 & 0 & 0 & 0 \end{pmatrix}, \quad h_{\text{Shrik}}^{(1)} = \begin{pmatrix} 2 & 1 & 0 & 1 \\ 1 & 1 & 0 & 0 \\ 0 & 0 & 0 & 0 \\ 1 & 0 & 0 & 1 \end{pmatrix}. \tag{16}$$

In the second iteration, let the hash function be $\text{Hash}(2, \{1,1,1,1,1,1\}) = 2$, $\text{Hash}(1, \{2,1,1,0,0,0\}) = 1$ and $\text{Hash}(0, \{1,1,0,0,0,0\})$. We can see that the refinement

process converges. Thus, both the top left corner node's representations are

$$h_{\text{Rook}} = h_{\text{Shirk}} = \text{Hash}(\{2, 1, 1, 1, 1, 1, 1, 0, 0, 0, 0, 0, 0, 0, 0, 0\}). \tag{17}$$

Thus Subgraph MPNNs cannot distinguish these two nodes. This is true regardless of which root node we choose. Therefore Subgraph MPNNs cannot distinguish the 4x4 Rook's graph and the Shirkhande graph.

### D.2 PROOF OF THEOREM 3.1

Using Lemma D.2, we can easily prove Theorem 3.1. Concretely, let $h_{i,j}$ be the number of 2-paths from node $i$ to node $j$, then $h_i = \frac{1}{2} \sum_{j \in N(i)} h_{i,j}$ is the number of 3-cycles involving node $i$, and $h_i = \sum_{j \in V} h_{i,j}$ is the number of 2-paths starting from node $i$. Similarly, let $h_{i,j}$ be the number of 3-paths from node $i$ to node $j$, then $h_i = \frac{1}{2} \sum_{j \in N(i)} h_{i,j}$ is the number of 4-cycles involving node $i$, and $h_i = \sum_{j \in N(i)} h_{i,j}$ is the number of 3-paths starting from node $i$. Note that we use summation over the neighbor of $i$, which can be implemented by marking the neighbor of $i$.

### D.3 PROOF OF THEOREM 3.2

To prove Theorem 3.2, consider the counter-example shown in figure 4.

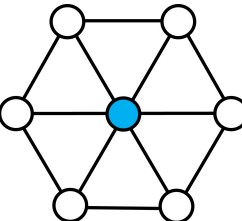 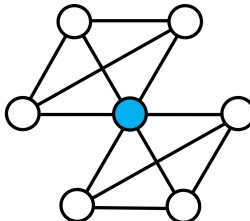

Figure 4: The two blue nodes in two graphs can not been distinguished by subgraph MPNNs. The left one is in six 5-cycles and six 4-paths while the right one is not.

Note that any Subgraph MPNNs get exactly the same representations for the two blue nodes. The left blue node involves in six 5-cycles and six 4-paths while the right one does not. Actually, one can construct similar counter-examples: one graph is a $2L$-cycle (white nodes) with all nodes connected to an additional node (blue node), another graph is two $L$-cycle (white nodes) with all nodes connected to an additional node (blue node). The first blue node involves in $(L + 2)$-cycles and $(L + 1)$-paths while the second one does not.

## E COUNTING POWER OF $I^2$-GNNs

In this section we prove Proposition 4.1, Theorems 4.1, 4.2 and 4.3. We first prove Lemma E.1, which is further used to prove Theorem 4.2 and part of Theorem 4.1.

For recap, $I^2$-GNNs assign two identifiers to root node $i$ and one of its neighbors $j$: $h_{i,j,k}^{(0)} = \mathbb{1}_{k=i} \oplus \mathbb{1}_{k=j}$. We assume the identity of $i$ and $j$ are always available in subgraph $G_{i,j}$.

**Lemma E.1.** $I^2$-GNNs can count 4-paths in forms of $(i \to j \to ... \to k)$ using $h_{i,j,k}$.

*Proof sketch.* The core idea stems from Lemma D.2, that using one unique identifier can count 3-paths. Recall that $I^2$-GNNs use two unique identifiers for two adjacent nodes $i$ and $j$, therefore one can show that $j$ can count number of 3-paths to another $k$ without passing node $i$. Equivalently, this is exactly the number of 4-paths from node $i$ to node $k$ while passing node $j$ in the first walk. $\square$

*Proof.* Consider the following message passing functions:

$$h_{i,j,k}^{(1)} = \mathbb{1}_{k \neq i} \sum_{l \in N(k)} \mathbb{1}_{l=j}, \tag{18a}$$

$$h_{i,j,k}^{(2)} = \mathbb{1}_{k \neq i} \mathbb{1}_{k \neq j} \sum_{l \in N(k)} h_{i,j,l}^{(1)}, \tag{18b}$$

$$h_{i,j,k}^{(3)} = \mathbb{1}_{k \neq i} \mathbb{1}_{k \neq j} \sum_{l \in N(k)} \left( h_{i,j,l}^{(2)} - h_{i,j,k}^{(1)} \right). \tag{18c}$$

One can see that $h_{i,j,k}^{(3)}$ is the number of 3-paths from $j$ to $k$ without passing $i$. Given that $i, j$ are adjacent, $h_{i,j,k}^{(3)}$ is equivalent to the number of 4-paths from $i$ to $k$ while passing $j$ in the first walk. □

### E.1 PROOF OF PROPOSITION 4.1

In this subsection we are going to show $I^2$-GNNs can distinguish the 4x4 Rook's graph and the Shrinkhande graph shown in Figure 3. Suppose we adopt 1-hop ego-network, then we find that for every root node $i$ of 4x4 Rook's graph, ego-net $G_i$ is the left graph shown in Figure 4 ($i$ is the blue node), and for every root node $i$ of Shrikhande graph, ego-net $G_i$ is the right graph shown in Figure 4 ($i$ is the blue node). Let $j$ be one of the neighbors of $i$. Now let us construct a message passing function to distinguish them. For instance, in the first layer we mark the common neighbors of $i$ and $j$, and in the second layer we determine the connectivity of these common neighbors.

$$\begin{pmatrix} h_{i,j,k,1}^{(1)} \\ h_{i,j,k,2}^{(1)} \end{pmatrix} = \mathbb{1}_{k \neq i} \mathbb{1}_{k \neq j} \sum_{l \in N(k)} \begin{pmatrix} \mathbb{1}_{l=i} \\ \mathbb{1}_{l=j} \end{pmatrix}, \tag{19a}$$

$$h_{i,j,k}^{(2)} = h_{i,j,k,1}^{(1)} h_{i,j,k,2}^{(1)} \sum_{l \in N(k)} h_{i,j,l,1}^{(1)} h_{i,j,l,2}^{(1)}. \tag{19b}$$

Then we can see that in 1-hop ego-net of the 4x4 Rook's graph, the node representations $h_{i,j,k}^{(2)}$ are all 0. In contrast, in 1-hop ego-net of the Shrikhande graph, the representations of common neighbors of $i$ and $j$ are 1. Therefore, $I^2$-GNNs can distinguish the 4x4 Rook's graph and the Shrikhande graph.

### E.2 PROOF OF THEOREM 4.2

$I^2$-GNNs can count 2-paths and 3-paths, since we have Theorem 3.1 and $I^2$-GNNs are more powerful than Subgraph MPNNs. Concerning 4-paths, according to Lemma E.1, we let $h_{i,j,k}$ be number of 4-paths in forms of $(i \to j \to ... \to k)$. Then one can see that $h_i = \sum_{j \in N(i)} \sum_{k \in V} h_{i,j,k}$ equals to number of 4-paths involving $i$.

### E.3 PROOF OF THEOREM 4.1

$I^2$-GNNs can count 3-cycles and 4-cycles since we have Theorem 3.1 and $I^2$-GNNs are more powerful than Subgraph MPNNs. Also, given Lemma E.1, we can let $h_{i,j,k}$ be the number of 4-paths in forms of $(i \to j \to * \to k)$. Then one can show that $h_{i,j} = \sum_{k \in N(i)} h_{i,j,k}$ equals to the number of 5-cycles involving edge $(i, j)$. Thus $h_i = \frac{1}{2} \sum_{j \in N(i)} h_{i,j}$ is exactly the number of 5-cycles involving node $i$.

Now let us prove that $I^2$-GNNs can count 6-cycles. We note that a necessary condition of a 6-cycle is that there exist a 4-path and a 2-path from node $i$ to $k$ simultaneously, as shown in left-most pattern in Figure 5. Thus let us consider the following function:

$$\#_0(i) = \sum_{k \in V} P_4(i, k) \cdot P_2(i, k), \tag{20}$$

where $P_2(i, k), P_4(i, k)$ are number of 2-paths and 4-paths from $i$ to $k$. However, $\#_0(i)$ is not equal to the number of 6-cycles: the right three patterns in Figure 5 also contribute to $\#_0(i)$. In fact, let

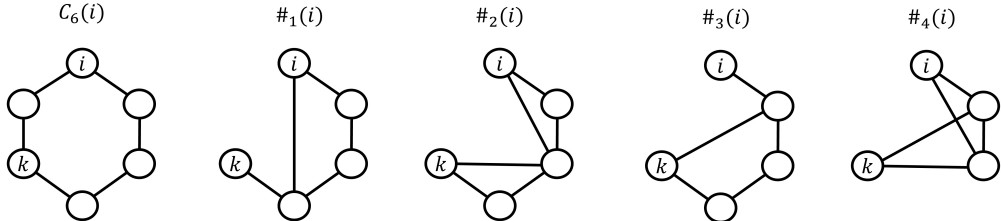

Figure 5: The first four patterns satisfy: there exist a 4-path and a 2-path from node $i$ to $k$ simultaneously.

$\#_1(i), \#_2(i), \#_3(i)$ be the number of these three patterns that involves node $i$, then we have the following relation (Fürer, 2017):

$$C_6(i) = \#_0(i) - \#_1(i) - \#_2(i) - \#_3(i). \tag{21}$$

We now prove that I²-GNNs can express $\#_0$ to $\#_3$, which consequently proves that I²-GNNs can count 6-cycles.

**Case 0**. Using Lemma E.1, we can let $h_{i,j,k} = P_4(i \to j \to ... \to k)$, i.e. the number of 4-paths in forms of $(i \to j \to ... \to k)$. Besides, we can let $h'_{i,j,k} = P_2(i,k)$, since counting 2-paths does not require identifier of $j$. Then we have

$$\sum_{j \in N(i)} \sum_{k \in V} h_{i,j,k} h'_{i,j,k} = \sum_{j \in N(i)} \sum_{k \in V} P_4(i \to j \to ... \to k) P_2(i,k)$$

$$= \sum_{k \in V} \left( \sum_{j \in N(i)} P_4(i \to j \to ... \to k) \right) P_2(i,k) \tag{22}$$

$$= \sum_{k \in V} P_4(i,k) P_2(i,k) = \#_0(i).$$

**Case 1.** To count $\#_1(i)$, notice that the 4-paths and 2-paths from $i$ to $k$ share the same second last node. This is equivalent to number of 4-paths with second last node being the neighbor of $i$. Therefore, we construct message passing functions as follows:

$$\begin{pmatrix} h_{i,j,k,1}^{(1)} \\ h_{i,j,k,2}^{(1)} \end{pmatrix} = \mathbb{1}_{k \neq i} \sum_{l \in N(k)} \begin{pmatrix} \mathbb{1}_{l=j} \\ \mathbb{1}_{l=i} \end{pmatrix}, \tag{23a}$$

$$h_{i,j,k}^{(2)} = \mathbb{1}_{k \neq i} \mathbb{1}_{k \neq j} \sum_{l \in N(k)} h_{i,j,l,1}^{(1)}, \tag{23b}$$

$$h_{i,j,k}^{(3)} = \mathbb{1}_{k \neq i} \mathbb{1}_{k \neq j} \sum_{j \in N(k)} h_{i,j,l,2}^{(1)} \left( h_{i,j,l}^{(2)} - h_{i,j,l,1}^{(1)} \right). \tag{23c}$$

Here $h_{i,j,k,2}^{(1)}$ is an indicator that labels the neighbor of $i$. Only the neighbor of $i$ can pass messages in the last step. Thus $h_{i,j,k}^{(3)}$ equals the number of 4-paths in forms of $(i \to j \to ... \to k)$ with the second last node being the neighbor of $i$. One can then see that $\sum_{j \in N(i)} \sum_{k \in V} h_{i,j,k}^{(3)} = \#_1(i)$.

**Case 2.** It is relatively difficult to count $\#_2(i)$. We first derive the following equation:

$$\#_2(i) = \left[ \sum_{k \in V} \sum_{j \in N(i) \cap N(k)} \left( C_3(i,j) - \mathbb{1}_{(i,k) \in E} \right) \left( C_3(j,k) - \mathbb{1}_{(i,k) \in E} \right) \right] - 2 \cdot \#_4(i), \tag{24}$$

where $C_3(i,j), C_3(j,k)$ are number of triangles that involve $(i,j)$ and $(j,k)$ as edges, and $\#_4(i)$ is the number of the rightest pattern shown in Figure 5. Let us explain why it is true. The term $\sum_{k \in V} \sum_{j \in N(i) \cap N(k)} C_3(i,j) C_3(j,k)$ counts the cases where $i$ and $k$ are both in triangles that

shares one common node. However, it is not equal to $\#_2(i)$, since (1) if $i, k$ are adjacent, then it contributes an additional triangles to each iterating node $j$ in the summation; (2) If rightest patterns in Figure 5 appears, it contributes two unexcepted counts. Thus by substracting these redundant counts, we obtain equation (24).

Now let us prove that I$^2$-GNNs can express $\#_2(i)$ through equation (24). We first transform the first term into:

$$\sum_{k \in V} \sum_{j \in N(i) \cap N(k)} \left( C_3(i,j) - \mathbb{1}_{(i,k) \in E} \right) \left( C_3(j,k) - \mathbb{1}_{(i,k) \in E} \right)$$

$$= \sum_{k \in V} \sum_{j \in N(i)} \mathbb{1}_{(j,k) \in E} \left( C_3(i,j) - \mathbb{1}_{(i,k) \in E} \right) \left( C_3(j,k) - \mathbb{1}_{(i,k) \in E} \right)$$

$$= \sum_{k \in V} \sum_{j \in N(i)} \mathbb{1}_{(j,k) \in E} C_3(i,j) \left( C_3(j,k) - \mathbb{1}_{(i,k) \in E} \right) - \mathbb{1}_{(j,k) \in E} \mathbb{1}_{(i,k) \in E} \left( C_3(j,k) - \mathbb{1}_{(i,k) \in E} \right)$$

$$= \sum_{j \in N(i)} C_3(i,j) \sum_{k \in V} \mathbb{1}_{k \in N(j)} \left( C_3(j,k) - \mathbb{1}_{k \in N(i)} \right) - \sum_{\substack{j \in N(i), \\ k \in V}} \mathbb{1}_{k \in N(i) \cap N(j)} \left( C_3(j,k) - \mathbb{1}_{k \in N(i)} \right). \tag{25}$$

We then use I$^2$-GNNs to encode $h_{i,j,k}$ as follows:

$$h_{i,j,k} = \left( \mathbb{1}_{k \in N(i)}, \quad \mathbb{1}_{k \in N(j)}, \quad C_3(j,k) \right)^\top \tag{26}$$

Note that these encodings are easily to expressed by message passing. Now we can finally express equation (25) using $h_{i,j,k}$:

$$\sum_{j \in N(i)} C_3(i,j) \sum_{k \in V} \mathbb{1}_{k \in N(j)} \left( C_3(j,k) - \mathbb{1}_{k \in N(i)} \right) - \sum_{\substack{j \in N(i), \\ k \in V}} \mathbb{1}_{k \in N(i) \cap N(j)} \left( C_3(j,k) - \mathbb{1}_{k \in N(i)} \right)$$

$$= \sum_{j \in N(i)} \left( \sum_{k \in V} h_{i,j,k,1} h_{i,j,k,2} \right) \left( \sum_{k \in V} h_{i,j,k,2}(h_{i,j,k,3} - h_{i,j,k,1}) \right)$$

$$- \sum_{j \in N(i)} \sum_{k \in V} h_{i,j,k,1} h_{i,j,k,2}(h_{i,j,k,3} - h_{i,j,k,1}). \tag{27}$$

Next let us prove that I$^2$-GNNs can count $\#_4(i)$. Image we label the neighbor of $i$ as $j$, then this pattern occurs if and only if there exists a node $k \in V \backslash \{i,j\}$ such that: (1) $k$ is the neighbor of $j$; (2) $k$ is the neighbor of the nodes in $N(i) \cap N(j)$. Thus we can write down the following message passing functions:

$$\begin{pmatrix} h_{i,j,k,1}^{(1)} \\ h_{i,j,k,2}^{(1)} \end{pmatrix} = \mathbb{1}_{k \neq i} \mathbb{1}_{k \neq j} \sum_{l \in N(k)} \begin{pmatrix} \mathbb{1}_{l=j} \\ \mathbb{1}_{l=i} \end{pmatrix}, \tag{28a}$$

$$h_{i,j,k}^{(2)} = \mathbb{1}_{k \neq i} \mathbb{1}_{k \neq j} h_{i,j,k,1}^{(1)} \sum_{l \in N(k)} h_{i,j,l,1}^{(1)} h_{i,j,l,2}^{(1)}. \tag{28b}$$

The resulting $\frac{1}{2} \sum_{j \in N(i)} \sum_{k \in V} h_{i,j,k}^{(2)}$ equals to $\#_4(i)$.

**Case 3.** Note that $\#_3(i)$ equals to the multiplication of the number 4-paths and 2-paths that share the same second node, i.e. $(i \rightarrow j \rightarrow * \rightarrow * \rightarrow *)$ and $(i \rightarrow j \rightarrow *)$. Consider the message passing defined as follows:

$$h_{i,j,k}^{(1)} = \mathbb{1}_{k \neq i} \sum_{l \in N(k)} \mathbb{1}_{l=j}, \tag{29a}$$

$$h_{i,j,k}^{(2)} = \mathbb{1}_{k \neq i} \mathbb{1}_{k \neq j} \sum_{l \in N(k)} h_{i,j,l}^{(1)}, \tag{29b}$$

$$h_{i,j,k}^{(3)} = \mathbb{1}_{k \neq i} \mathbb{1}_{k \neq j} h_{i,j,k}^{(1)} \sum_{l \in N(k)} \left( h_{i,j,l}^{(2)} - h_{i,j,k}^{(1)} \right). \tag{29c}$$

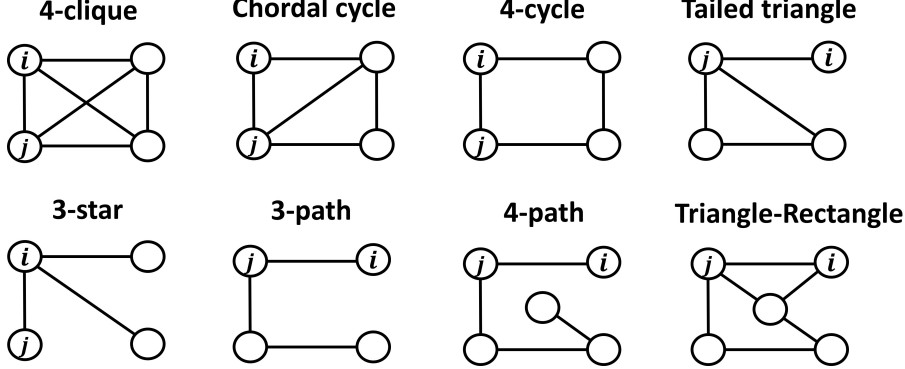

Figure 6: Some of the connected Graphlets with size 4 and 5. We define the node-level counting $C(\text{graphlet}, i, G)$ be the number of graphlets involving $i$ at the position shown in the figure.

Then $\#_3(i) = \sum_{j \in N(i)} \sum_{k \in V} h_{i,j,k}^{(3)}$.

Since we can express $\#_0(i), \#_1(i), \#_2(i), \#_3(i)$, we are able to express number of 6-cycles involving node $i$ using $C_6(i) = \#_0(i) - \#_1(i) - \#_2(i) - \#_3(i)$.

### E.4   PROOF OF THEOREM 4.3

All connected graphlets with size 3 include 2-paths and 3-cycles, which have been proven before. Now let us consider all connected graphlets with size 4, shown by the first six graphs in Figure 6.

**4-Cliques.** The definition of 4-clique is equivalent to a triangle with all nodes connected to an additional node. Thus if $i, j$ are the root node and one of its neighbors, then number of 4-cliques equals to number of triangles that involve $j$ but not $i$ and whose nodes are connected to $i$. Based on this observation, we design the following message passing:

$$\begin{pmatrix} h_{i,j,k,1}^{(1)} \\ h_{i,j,k,2}^{(1)} \end{pmatrix} = \mathbb{1}_{k \neq i} \mathbb{1}_{k \neq j} \sum_{l \in N(k)} \begin{pmatrix} \mathbb{1}_{l=i} \\ \mathbb{1}_{l=j} \end{pmatrix}, \tag{30a}$$

$$h_{i,j,k}^{(2)} = \mathbb{1}_{k \neq i} \mathbb{1}_{k \neq j} h_{i,j,k,1}^{(1)} h_{i,j,k,2}^{(2)} \sum_{l \in N(k)} h_{i,j,l,1}^{(1)} h_{i,j,l,2}^{(1)}. \tag{30b}$$

Then on can find that $\frac{1}{6} \sum_{j \in N(i)} \sum_{k \in V} h_{i,j,k}^{(2)}$ equals to the number of 4-clique involving node $i$.

**Chordal cycles.** We first mark all the common neighbors of $i, j$ by one layer of message passing, represented by $h_{i,j,k}^{(1)} = \mathbb{1}_{k \in N(i)} \mathbb{1}_{k \in N(j)}$. Then the key is to find number of triangles that involves both the common neighbors and $j$. We therefore construct the second layer message passing as:

$$h_{i,j,k}^{(2)} = \mathbb{1}_{k \neq i} \mathbb{1}_{k \neq j} \sum_{l \in N(k)} h_{i,j,l}^{(1)} \tag{31}$$

Then one should see that $\frac{1}{4} \sum_{i \in V} \sum_{j \in N(i)} \sum_{k \in N(j)} h_{i,j,k}^{(2)}$ equals to the number of chordal cycles in the graph.

**Tailed triangles.** Let $i$ be the node that connects tailed node $j$. Then let $h_{i,j}$ be number of triangles involving $i$ but not $j$. One can see that $\sum_{i \in V} \sum_{j \in N(i)} h_{i,j}$ is the number of tailed triangles.

**4-cycles and 3-paths.** By theorems 4.1 and 4.2.

### E.5   SOME OTHER GRAPHLETS

**4-paths.** By theorem 4.2.

**Triangle-Rectangles.** Let $i$ be the node that is in the triangle but not in the rectangle. Let $j$ be the neighbor of $i$. Recall that we can use the labeling of $j$ to mark the 3-hop nodes. Then by determining if these nodes are the common neighbors of $i, j$, we can count the number of triangle-rectangle.

# F   ADDITIONAL DETAILS OF THE NUMERICAL EXPERIMENTS

Table 7: Statistics of the synthetic, ChEMBL, QM9, ZINC and OGB datasets.

| Dataset | #Graphs | Avg. #Nodes | Avg. #Edges | Task type |
|---------|---------|-------------|-------------|-----------|
| Synthetic | 5,000 | 18.8 | 31.3 | Node regression |
| ChEMBL | 16,200 | 21.5 | 22.9 | Node regression |
| QM9 | 130,831 | 18.0 | 18.7 | Graph regression |
| ZINC-12k | 12,000 | 23.2 | 24.9 | Graph regression |
| ZINC-250k | 249,456 | 23.1 | 24.9 | Graph regression |
| ogbg-molhiv | 41,127 | 25.5 | 27.5 | Graph classification |

## F.1   GRAPH STRUCTURES COUNTING

**Dataset.** The synthetic dataset is provided by open-source code of GNNAK on github. The real-world dataset ChEMBL (https://www.ebi.ac.uk/chembl/) contains bioactive molecules with drug-like properties. We filter the dataset by the following process: we compute the average number of 4,5,6-cycles per atom for each molecule, add uniform noises ($U(-0.05, 0.05)$) and sort the molecules based on the average number of cycles per node. We then apply CD-HIT (Li & Godzik, 2006), a clustering algorithm, to cluster the molecules with similarity measured by their fingerprints. We screen out 16,200 cluster centers such that each pair of them has a similarity less than 0.4. These 16,200 centers (molecules) are used to build our final dataset. The ground-truth labels are obtained by networkx.algorithms.isomorphism package.

**Models.** For ID-GNNs, Nested GNNs and $I^2$-GNNs, we uniformly adopt a 5-layer GatedGNNs (Li et al., 2015) as base GNNs. For GNNAK+, we use 5 GNNAK+ layers, each of which is a 1-layer GatedGNN. The embedding size of mentioned models above is 64. The subgraph height is 1, 2, 2, 3 for 3, 4, 5 and 6-cycles respectively. The initial features are augmented with shortest path distances except ID-GNNs. For PPGNs, we use 4 PPGN layers with 2 blocks and embedding size 300. The subgraph height $h$ is 1,2,2,3,2,2,1,4,2 for 3-cycles, 4-cycles, 5-cycles, 6-cycles, tailed triangles, chordal cycles, 4-cliques, 4-paths and triangle-rectangle respectively, so that the subgraph can include the graphlet.

**Training details.** The training/validation/test splitting ratio is 0.3/0.2/0.5 on synthetic dataset, and 0.6/0.2/0.2 on ChEMBL dataset. We uniformly use Mean Absolute Error as loss function except 3-cycle. In the 3-cycle case we use Mean Square Error instead, as some training loss get stuck. We use Adam optimizer with initial learning rate 0.001, and use plateau scheduler with patience 10 and decay factor 0.9. We train 2,000 epochs for each models. The batch size is 256.

## F.2   MOLECULAR PROPERTIES PREDICTION

### F.2.1   QM9

**Dataset.** The QM9 dataset is provided by pytorch geometric.

**Models.** We adopt 5-layer NNConv (Gilmer et al., 2017) as base GNNs of $I^2$-GNNs. The embedding size is 64, and the subgraph height is 3. The initial features are augmented with shortest path distances and resistance distances (Lü & Zhou, 2011).

**Training details.** The training/validation/test splitting ratio is 0.8/0.1/0.1. We use Adam optimizer with initial learning rate 0.001, and use plateau scheduler with patience 10 and decay factor 0.95. We train 400 epochs with batch size 64 for each target separately.

### F.2.2   ZINC

**Dataset.** We use the dataset provided by Dwivedi et al. (2020). It contains ZINC-12k and a ZINC-250k.

**Models.** We adopt 5-layer GINE (Hu et al., 2019) as base GNNs of I$^2$-GNNs. The embedding size is 64, and the subgraph height is 3. The initial features are augmented with shortest path distances and resistance distances. To further improve the empirical performance, we concatenate the pooled node embedding $h_k$ to the corresponding node $h_{i,j,k}$ at each layer: $h_{i,j,k} \leftarrow h_{i,j,k} \oplus h_k$.

**Training details.** The training/validation/test splitting is given by the dataset. We use Adam optimizer with initial learning rate 0.001, and use plateau scheduler with patience 10 and decay factor 0.95. We train 1,000 epochs with batch size 256 on Zinc-12k, and train 800 epochs with batch size 256 on Zinc-250k.

### F.2.3   OGB

**Dataset.** The ogbg-molhiv dataset is provided by Open Graph Benchmark (OGB).

**Models.** We adopt 5-layer GINE as base GNNs of I$^2$-GNNs/ The embdding size is 300, and the subgraph height is 4. The initial node features are augmented with shortest path distances and resistance distances.

**Training details.** The training/validation/test splitting is given by the dataset. We use Adam optimizer with initial learning rate 0.001, and use step scheduler with step size 20 and decay factor 0.5. We train 50 epochs with batch size 64.

## G   CYCLES STATISTICS ON DATASETS

We demonstrate the the average number of cycles on different datasets. We can see that 5-cycles and 6-cycles are dominant compared to 3-cycles and 4-cycles. It reflects the importance of counting to 6-cycles.

Table 8: Average number of cycles per graph on the synthetic, ZINC and OGB datasets.

| Dataset | Avg. # 3-Cycles | Avg. # 4-Cycles | Avg. # 5-Cycles | Avg.# 6-Cycles |
|---|---|---|---|---|
| Synthetic | 5.04 | 10.60 | 21.67 | 41.6 |
| ChEMBL | 0.06 | 0.03 | 0.69 | 1.72 |
| ZINC-12k | 0.06 | 0.02 | 0.85 | 1.80 |
| ogbg-molhiv | 0.03 | 0.04 | 0.70 | 2.28 |

## H   MORE ABOUT COUNTING EXPERIMENTS

### H.1   COUNTING ERROR VERSUS NUMBER OF CYCLES

In addition, we try to understand the performance of I$^2$-GNNs by further analysis and visualization. In subfigures (a,c) in Figure 7 we visualize the label distribution of 5-cycles and 6-cycles on synthetic dataset, i.e. nodes with different number of cycles. We see that the number of nodes with increasing cycles decays exponentially. Thus it implies the difficulty to generalize to nodes with more cycles. In the subfigures (b,d) in Figure 7 we show the corresponding prediction average MAE on nodes with different number of cycles. we can see that the MAE of Subgraph GNNs increases dramatically as the number of cycles become larger. In contrast, I$^2$-GNNs have a almost flatten curve, implying it fits the real counting function well and thus generalizes better.

### H.2   RESULTS OF COUNTING AT GRAPH LEVEL

Using the same setup in node-level counting, we also conduct counting experiments at graph level. Note that I$^2$-GNNs can count 6-cycles at graph level with theoretical guarantees (by Theorem 4.1), while other Subgraph GNNs are unclear. Interestingly, the experimental results in Table 9 show that all Subgraph GNNs attain comparable performance with respect to I$^2$-GNNs and PPGNs. It may be because: (1) they capture some statistical biases that are highly correlated to graph-level counting functions; (2) they can count up to 6-cycles at graph level theoretically. The graph-level counting power of Subgraph GNNs is unclear from 5-cycles to 7-cycles and worthy exploring in the future research.

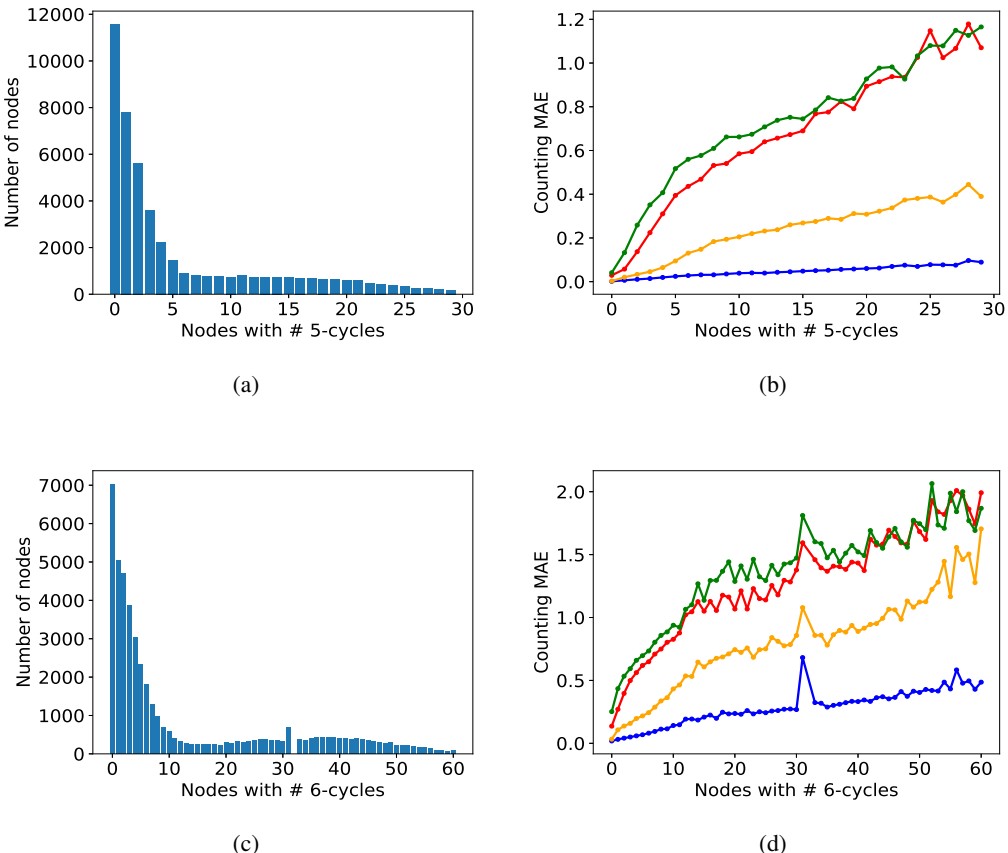

Figure 7: Left column: distribution of number of cycles of nodes. Right column: prediction MAE conditioning on nodes with certain number of cycles.

Table 9: Normalized MAE results with errors of counting substructures at graph level. The reported uncertainty is one times standard deviation.

| Method | Synthetic (norm. MAE) | | | |
|---|---|---|---|---|
| | 3-Cycle | 4-Cycle | 5-Cycle | 6-Cycle |
| GatedGCN | $0.3845\pm0.0057$ | $0.2481\pm0.0010$ | $0.1849\pm0.0014$ | $0.1500\pm0.0036$ |
| ID-GNN | $0.0008\pm0.0005$ | $0.0064\pm0.0001$ | $0.0070\pm0.0010$ | $0.0182\pm0.0010$ |
| NGNN | $0.0009\pm0.0002$ | $0.0054\pm0.0005$ | $0.0077\pm0.0026$ | $0.0249\pm0.0010$ |
| GNNAK+ | $0.0021\pm0.0004$ | $0.0177\pm0.0026$ | $0.0233\pm0.0022$ | $0.0412\pm0.0025$ |
| PPGN | $0.0013\pm0.0007$ | $0.0050\pm0.0005$ | $0.0123\pm0.0013$ | $0.0159\pm0.0015$ |
| $I^2$-GNN | $0.0010\pm0.0001$ | $0.0066\pm0.0011$ | $0.0098\pm0.0005$ | $0.0237\pm0.0011$ |

## I ABLATION STUDY

In ablation study, we focus on studying the impact of the key element of $I^2$-GNNs: the additional identifiers. Concretely, let $i$ be the root node and $j$ be the branching node (neighbor of $i$), we drop all the node labeling concerning $j$ in the subgraph $G_{i,j}$. It results in a model denoted $I^2$-GNNs (single) with only one identifier and all other hyper-parameters unchanged. In additional, we also study the effect of the shortest path labeling v.s. identity labeling. We implement $I^2$-GNNs (id), where the shortest path labelings $z_{i,j,k} = \mathrm{spd}(i,k) \oplus \mathrm{spd}(j,k)$ are replaced with the identity labeling $z_{i,j,k} = \mathbb{1}_{i=k} \oplus \mathbb{1}_{j=k}$. From Table 10 we can see that performance drops greatly for $5, 6$-cycles,

Table 10: Ablation study on single node labeling and identity labeling. The reported uncertainty is one times standard deviation.

| Dataset/Task | I²-GNNs | I²-GNNs (single) | I²-GNNs (id) |
|---|---|---|---|
| 3-Cycle | 0.0003±0.0000 | 0.0004±0.0002 | 0.0003±0.0001 |
| 4-Cycle | 0.0015±0.0001 | 0.0021±0.0001 | 0.0015±0.0001 |
| 5-Cycle | 0.0028±0.0001 | 0.0426±0.0005 | 0.0034±0.0000 |
| 6-Cycle | 0.0078±0.0007 | 0.0465±0.0009 | 0.0103±0.0021 |
| 4-Clqiue | 0.0003±0.0001 | 0.0604±0.0902 | 0.0012±0.0014 |
| 4-Path | 0.0041±0.0010 | 0.0241±0.0007 | 0.0019±0.0001 |
| Triangle-Rectangle | 0.0026± 0.0002 | 0.0566±0.0015 | 0.0017±0.0001 |

4-cliques, 4-paths and triangle-rectangle after removing the additional identifiers. This is consistent to our claim that the additional identifiers do boost the counting power. In the comparison between shortest path labeling and identity labeling, we can see that they do not dominate each other. This is because they have the same representational power and thus the results should depend on the inductive bias of the task.

## J  DISCUSSION ON COMPLEXITY

### J.1  COMPARISON TO OTHER GNN MODELS

I²-GNNs increase the representational power by extending Subgraph MPNNs (single identifier) to a pair of identifiers. In the language of equivariant tensors, I²-GNNs lift the original graph (adjacency matrix, 2-order tensor) to a 4-order tensor (first two indices for the two identifiers, last two indices for adjacency), and perform message passing on this 4-order tensor. Thus one should expect I²-GNNs' power to be upper bounded by 4-IGN/4-WL, as implied in Frasca et al. (2022). Regarding the complexity, suppose a graph has $N$ nodes with average node degree $d$, and the maximum subgraph size is set to $s$. One layer message passing takes $\mathcal{O}(d)$ time for one node. By leveraging the sparsity of graphs, I²-GNNs reach a linear time complexity $\mathcal{O}(Nsd^2)$ w.r.t. node number $N$. Specifically, counting 6-cycles only requires extracting 3-hop rooted subgraphs, resulting in $\mathcal{O}(Nd^5)$ complexity. Note that in molecule datasets the node degree is usually small (e.g., QM9 has an average node degree 2.1), which makes the computation feasible. This is in contrast to the at least $\mathcal{O}(N^3)$ complexity of those high-order GNNs such as PPGNs (Maron et al., 2019a) and $k$-IGNs (Maron et al., 2018). Some works also try to design localized high-order GNNs. For example, $k$-GNNs (Morris et al., 2019) and $\delta$-$k$-LGNNs (Morris et al., 2020b) reduce the time complexity to linear by constraining the $k$-tuples to be *connected* and aggregating messages from neighbors only. Compared to them, I²-GNNs have an exclusive advantage: the initial features $x_{i,j,k}$ can be augmented with some three-tuple attributes, such as $x_{i,j,k} = (\mathrm{spd}(i,k), \mathrm{spd}(j,k))$ or other structural/positional encodings. Besides, I²-GNNs have provable substructure counting power. Local relational pooling (Chen et al., 2020) is a universal approximator for permutation invariant functions over subgraphs, but the $\mathcal{O}(Ns!)$ complexity is almost infeasible. A contemporary work (Qian et al., 2022) also discusses the possibility of high-order Subgraph GNNs: they propose to tie each subgraph with a $k$-tuple. However, their main focus is the theoretical comparison with $k$-WL test. Besides, the complexity of their original model grows exponentially.

Table 11: Space and time complexity of different GNN models.

| Model | MPNN | Subgraph MPNN | I²-GNN | PPGN | 1-2-3 GNN |
|---|---|---|---|---|---|
| Space complexity | $\mathcal{O}(N)$ | $\mathcal{O}(Ns)$ | $\mathcal{O}(Nsd)$ | $\mathcal{O}(N^2)$ | $\mathcal{O}(Nd^2)$ |
| Time complexity | $\mathcal{O}(Nd)$ | $\mathcal{O}(Nsd)$ | $\mathcal{O}(Nsd^2)$ | $\mathcal{O}(N^3)$ | $\mathcal{O}(Nd^3)$ |

### J.2  EMPIRICAL COMPLEXITY EVALUATION

We empirically compare the models' complexity through: (1) number of parameters (#Parms); (2) Maximal GPU memory usage (Max. memory usage) and (3) inference time. We adopt node-level counting task. The model implementation is exactly the same as in F.1, except that PPGN's embed-

Table 12: Empirical evaluation of complexity of I$^2$-GNNs, Subgraph GNNs and PPGNs.

| Model | MPNN | ID-GNN | NGNN | GNNAK+ | I$^2$-GNN | PPGN |
|---|---|---|---|---|---|---|
| #Parms | 27k | 102k | 127k | 251k | 143k | 96k |
| Memory usage (GB) | 1.88 | 2.35 | 2.34 | 2.35 | 3.59 | 2.30 |
| Inference time (ms) | 2.10 | 5.73 | 6.03 | 16.07 | 20.62 | 35.33 |

ding size is adjusted to 64 for a fair comparison. Batch size is 256, and we run 10 epochs to compute the average inference time per batch and the maximal memoery usage during inference. We use thop package to estimate the number of parameters. Note that in the realizations of listed Subgraph GNNs and I$^2$-GNNs, the subgraph extraction is preprocessed and thus the forward computation is in parallel. This causes a greater memory usage but a faster speed of training/inference.

From the Table 12 we can see that the inference time of I$^2$-GNNs is feasible, since the average node degree of counting dataset is approximately 3.3.

## K  SCALING UP: BRANCH NODE SAMPLING

Like Subgraph MPNNs, the extracted $K$-hop ego-networks size in I$^2$-GNNs is $\mathcal{O}(d^K)$. In addition, compared to Subgraph MPNNs, I$^2$-GNNs further increase the complexity by a factor of node degree $d$. In case where average node degree is large or the node degree distribution is heavy-tailed, the computational cost might still be unfordable.

Table 13: Statistics of the selected datasets from TUD benchmark.

| Dataset | #Graphs | Avg. #Nodes | Avg. #Edges |
|---|---|---|---|
| ENZYMES | 600 | 32.6 | 62.1 |
| IMDB-BINARY | 1,000 | 19.8 | 96.5 |

One can apply any subgraph sampling strategy to address the growing subgraph size problem, as the one in Zhao et al. (2021). Here we focus on alleviate the growing branch node problem. We provide a simple but effective sampling strategy: randomly sample a fixed number of branch nodes to bound the additional factor by a constant. The sampling are only applied to training process, and thus permutational equivariance still holds during test.

Table 14: Test accuracy (%) with one standard deviation on TUD benchmark. I$^2$-GNN-sample refers to I$^2$-GNNs that randomly choose and fix two branch nodes to label before training.

| Datasets | Base GNN | Nested GNN | I$^2$-GNN | I$^2$-GNN-sample |
|---|---|---|---|---|
| ENZYMES | 27.3±7.8 | 31.7±3.7 | 35.8±7.1 | 33.8±5.9 |
| IMDB-BINARY | 70.2±5.1 | 71.4±5.9 | 73.5±3.0 | 72.7±3.9 |

Table 15: #parameters / training time per 200 epochs (s) / inference time per epoch (ms) on TUD benchmark. Training/inference time are estimated on full dataset.

| Datasets | Base GNN | Nested GNN | I$^2$-GNN | I$^2$-sample-GNN |
|---|---|---|---|---|
| ENZYMES | 24k/19.45/59.09 | 25k/30.31/107.44 | 34k/62.82/213.44 | 34k/45.68/213.03 |
| IMDB-BINARY | 24k/31.19/88.49 | 25k/65.28/238.91 | 34k/355.98/1273.30 | 34k/**100.33**/1270.62 |

Formally, let $N(i)$ be the neighbor of root node $i$. An I$^2$-sample-GNN has the same architecture with I$^2$-GNN during training, except that

- in equation (6), the branch node $j$ iterates over $\text{Sample}_S(N(i))$ instead of $N(i)$, where $\text{Sample}_S$ is a sampling function that randomly chooses $S$ elements from the multiset.
- in equation (8), the node read-out function becomes

$$\forall i \in V, \quad h_i = \alpha_i \cdot R_{\text{node}}(\{h_{i,j} | j \in \text{Sample}_S(N(i))\}), \tag{32}$$

  where $\alpha_i = \frac{d_i}{S}$ if $R_{\text{node}}$ is additive to node degree $d_i$ (such as sum pooling), and $\alpha_i = 1$ if $R_{\text{node}}$ is independent of node degree $d_i$ (such as average pooling).

On the other hand, sampling is disabled when doing inference in order to assure permutational equivariance, i.e. $I^2$-sample-GNN use the exactly same architecture with $I^2$-GNN during inference. Note that because of the stochastic branch node sampling, $I^2$-sample-GNN would not be able to learn exact counting function. Instead it approximates the counting function in an expectational meaning.

To test the effectiveness of branch node sampling, we choose some high-node-degree datasets from TUD benchmark (Morris et al., 2020a). Table 13 above shows the dataset statics. We randomly choose and fix two branch nodes for each root node before training. The sampling does not apply to testing. We compare performance of $I^2$-GNNs with branch node sampling to that of original $I^2$-GNNs, base GNN (4-layer GraphConv) and Nested GNN. The subgraph depth is 3 for ENZYMES and 2 for IMDB-BINARY. We train each model for 200 epochs, with initial learning rate 0.002 and decay rate 0.5 per 50 epochs. We report the 10-fold cross validation results in Table 14. It demonstrates that $I^2$-GNNs with even simple branch node sampling still can bring improvements to baselines and achieve competitive performance to original $I^2$-GNNs.

Moreover, we compare the number of parameters, the training/test time for these models, as shown in Table 15. As we expected, $I^2$-sample-GNN can significantly reduce the training time, especially for high-node-degree dataset IMDB-BINARY.

