# OpenReview forum: "Boosting the Cycle Counting Power of Graph Neural Networks with I$^2$-GNNs"
_ICLR.cc/2023/Conference — ICLR 2023 poster_

### Official Review · Reviewer_Yf1m · 2022-10-21

**Confidence:** 4
**Correctness:** 3
**Technical Novelty And Significance:** 4
**Empirical Novelty And Significance:** 3
**Recommendation:** 8

**Clarity, Quality, Novelty And Reproducibility:**

\
**Clarity**

The paper is generally clearly written.

Explaining what the accuracies mean in Table 1 and their extreme differences ($0\%$ vs. $100\%$) in Table 1 would improve the clarity (especially for non-expert readers).

\
**Quality**

The paper is technically sound and both the theory and the experiments support the claims.

Since the complexity depends on the node degrees, an inference time and number of parameter comparison on real data (e.g., datasets of tables 4 and 5) would strengthen the quality of the paper.

Since benzene rings were used as an example real-world scenario of counting power, performing experiments (or highlighting if already performed) on real data with benzene rings would also strengthen the soundness of the claims.


\
**Novelty**

The proposed model is the first linear-time GNN model to count $6$-cycles.

The novelty of the work can be further strengthened by positioning the contributions with respect to GNNs for link prediction which also exploit subgraphs around each edge.

A discussion of the differences between the expressivity of labelling trick vs. I$^2$-GNNs would further strengthen the contributions.

1. Labelling Trick: A Theory of Using Graph Neural Networks for Multi-Node Representation Learning, In NeurIPS'21
2. Link Prediction Based on Graph Neural Networks, In NeurIPS'18



\
**Reproducibility**

The main part and the supplementary part include enough material, e.g., proofs of theorems, dataset details, baselines with references, hyperparameters, ablation study, for an expert to replicate the results of the paper.

___

**Strength And Weaknesses:**

\
**Strengths**

\+ The paper is well-organised, well-written, and the problem of counting power is well-motivated.

\+ The proposed method is potentially scalable to large-scale real-world data and potentially useful for real-world chemical data involving cycles of length at most $6$ (e.g., benzene rings).

\+ Under certain conditions (e.g., bounded node degree), the proposed model is the first linear-time GNN model to count $6$-cycles.


**Weakness**

The only major weakness of the paper is the lack of positioning/comparison of the idea of using multiple identifiers for node pairs (of edges) with labelling tricks [1] (used in the context of link prediction)

[1] Labelling Trick: A Theory of Using Graph Neural Networks for Multi-Node Representation Learning, In NeurIPS'21

___

**Summary Of The Paper:**

Message-passing neural networks (MPNNs) and their extensions (Subgraph MPNNs) have inspired state-of-the-art models for graph data; however, their counting power is still limited and it is known that Subgraph GNNs cannot count cycles with more than $4$ nodes.

To boost the counting power, this paper extends the idea of having a single root node identifier in Subgraph GNNs to two identifiers for a (root node, neighbour) pair resulting in the proposed I$^2$-GNN model.

I$^2$-GNN has the following properties:
1. It can count all cycles of length at most $6$,
2. It has linear space and time complexity provided the node degree is bounded,
3. It achieves competitive results on open benchmarks compared to existing models.

___

**Summary Of The Review:**

While the paper is well-motivated and the claims are well-supported with theory and experiments, positioning with respect to relevant prior work can further strengthen the contributions.
___

---

> ### Author Response · Authors · 2022-11-17
> **Author response**
>
> We thank the reviewer for their positive review.
>
> > **Q1:** ``The only major weakness of the paper is the lack of positioning/comparison of the idea of using multiple identifiers for node pairs (of edges) with labelling tricks [1] (used in the context of link prediction)."
>
> **A1:** We thank the reviewer for the suggestion. We add extra discussion about labeling trick into the manuscript. Indeed, both I$^2$-GNNs and labeling trick use multi-node labeling. But from a high-level view, labeling trick is a method that focuses on representing **multi-node**. In contrast, I$^2$-GNNs' ultimate goal is to learn **graph** or **single-node** representations by leveraging subgraph representations. As a result, from a technical point of view, labeling trick only labels the target 2-tuple for downstream link prediction, while I$^2$-GNNs traverse and label every connected 2-tuples to merge into the final node/graph representation.
>
> > **Q2:** ``Since the complexity depends on the node degrees, an inference time and number of parameter comparison on real data (e.g., datasets of tables 4 and 5) would strengthen the quality of the paper."
>
> **A2:** We thank the reviewer for the suggestion. We compare the training time, inference time and number of parameters between different models on TUD benchmark. Please kindly check the result in Table 2 in our response ``Main revisions to the paper" on top.
>
> [1] Labelling Trick: A Theory of Using Graph Neural Networks for Multi-Node Representation Learning, In NeurIPS'21

---

### Official Review · Reviewer_Mx4a · 2022-10-24

**Confidence:** 3
**Correctness:** 4
**Technical Novelty And Significance:** 3
**Empirical Novelty And Significance:** 2
**Recommendation:** 5

**Clarity, Quality, Novelty And Reproducibility:**

Clarity: The paper is well written, and the idea is clear and well presented.\
Quality: The quality of the paper is high.\
Novelty: The paper has some novelty, as it enhances Subgraph MPNNs with a new strategy of node pair identifiers, but the general concept of node identifiers in GNNs has been studied in previous works. Also, a more detailed related work, discussing these approaches is missing.\
Reproducibility: The experimental setup is well described and also the source code is provided by the authors.


**Strength And Weaknesses:**

Strengths:
- The authors of the paper present some new theoretical results regarding Subgraph MPNNs. Specifically, they prove that the proposed models can count 3-cycles and 4-cycles, but cannot count 5-cycles or any longer cycles.

- The authors also present a new Subgraph MPNN architecture that is more powerful than existing Subgraph MPNNs. They also prove that it can count all cycles with length less than 7 which is important for real-world applications (e.g., identifying benzene rings).

- The proposed models achieve competitive performance in graph classification and regressions tasks (QM9, ZINC and ogbg-molhiv) and demonstrate increased expressiveness in discriminating non-isomorphic graphs and in substructure counting.

Weaknesses:
- My main concern with this paper is with regards to the novelty of the work, as in essence, it proposes to add an extra identifier for each pair of nodes. There are many works that add unique node identifiers in GNNs to increase the expressive power of the models [1,2] and in the current paper, a proper discussion of these approaches is missing. Also, a comparison with [3,4] would be interesting, as they also build more expressive GNNs with extra node features.

- The proposed model generates a new subgraph for each edge in the original graph, and therefore the computational complexity is higher than that of other subgraph MPNNs. As the authors mentioned, the proposed model is evaluated only on graphs with a small average node degree. I would suggest the authors discuss how the proposed approach can be applied to graphs with high average node degree and also evaluate their model on such graphs, for example, in the TUD benchmark [5].

[1] Andreas Loukas. How hard is to distinguish graphs with graph neural networks? In Advances in Neural Information and Processing Systems (NeurIPS), 2020.\
[2] Andreas Loukas. What graph neural networks cannot learn: depth vs width. In ICLR, 2020.\
[3] Dwivedi, Vijay Prakash, et al. "Graph neural networks with learnable structural and positional representations." arXiv preprint arXiv:2110.07875 (2021).\
[4] Wijesinghe, Asiri, and Qing Wang. "A New Perspective on" How Graph Neural Networks Go Beyond Weisfeiler-Lehman?"." International Conference on Learning Representations. 2021.\
[5] Morris, Christopher, et al. "Tudataset: A collection of benchmark datasets for learning with graphs." arXiv preprint arXiv:2007.08663 (2020).

**Summary Of The Paper:**

The authors of this paper propose a new Subgraph MPNN architecture, called $I^2$-GNNs, that can count up to 6-cycles in contrast to other Subgraph MPNNs which cannot count such long cycles. For each rooted subgraph centered at node $i$, they construct d(i) subgraphs, where d(i) is the degree of the center node, defining a mapping between the generated subgraphs and the neighbors of the root node. Then, they assign a unique identifier to the center node $i$, and another unique identifier to one of the neighbors of the center node, for each generated subgraph. They use MPNNs in each subgraph to construct the graph's final representation, followed by a readout layer. They evaluate their model in cycle counting tasks as well as in molecular property prediction benchmarks, achieving competitive performance.


**Summary Of The Review:**

The proposed method is technically sound, well written, and leads to performance gains on some molecular property prediction datasets. However, I think that the main idea of incorporating identity node features has been studied in the past thoroughly and the paper does not discuss these methods in details. Specifically, I would suggest the authors do the following:
- Discuss more and extend the related work with other similar approaches that incorporate extra node features/identifiers and explain why the current approach is more suitable than the others.
- Compare with more baselines and evaluate the model in more benchmarks (see above).
- Explain how the current approach can be applied to dense graphs, where the average node degree is high, probably by employing some sampling strategies.

I would be happy to increase my score if my concerns are well addressed.

---

> ### Author Response · Authors · 2022-11-17
> **Author response**
>
> We thank the reviewer for their comprehensive and constructive review.
>
> > **Q1:** ``My main concern with this paper is with regards to the novelty of the work, as in essence, it proposes to add an extra identifier for each pair of nodes. There are many works that add unique node identifiers in GNNs to increase the expressive power of the models [1,2] and in the current paper, a proper discussion of these approaches is missing. Also, a comparison with [3,4] would be interesting, as they also build more expressive GNNs with extra node features."
>
> **A1:** Thank you for introducing these relevant works. We add discussion on them in the revised paper. It is true that both our work and [1, 2] use unique node identifiers to augment expressive power, but the ways how they use node identifiers are different in essence. For example, [1, 2] assume **each** node has a unique identifier in the **original graph**, and perform message passing on this original graph. The permutation equivariance is learned from data and generally does not hold. In contrast, I$^2$-GNN is a Subgraph GNN that assigns unique identifiers to the root node and the branch node in a **separate subgraph**, and all the other nodes in the subgraph keep **anonymous**. By iteratively assigning root nodes and branch nodes, I$^2$-GNNs preserve permutation equivariance by design.
>
> Concerning [3, 4], [3] introduces an extra node feature channel for positional encodings, while [4] proposes a way to compute the similarity $\omega_{i,j}$ between a node $i$ and its neighbor $j$ and uses it to weight the incoming messages $m_{i,j}$. The extra model capacity highly relies on the design of these heuristics. Again, the key of I$^2$-GNNs is to **partially** label some nodes and keep the others anonymous. Its expressive power comes from distinguishing some of the nodes from others, and any type of extra node features can fit in (node identifiers, distance encodings, positional encodings, etc.).
>
> > **Q2:** ``The proposed model generates a new subgraph for each edge in the original graph, and therefore the computational complexity is higher than that of other subgraph MPNNs. As the authors mentioned, the proposed model is evaluated only on graphs with a small average node degree. I would suggest the authors discuss how the proposed approach can be applied to graphs with high average node degree and also evaluate their model on such graphs, for example, in the TUD benchmark."
>
> **A2:** We thank the reviewer for the suggestion. How to further improve the scalability is indeed a practically important problem, especially for graphs with high node degree. Note that a high node degree $d$ increases the complexity in two main aspects: (1) the subgraph size, if using $k$-hop ego-networks, is $\mathcal{O}(d^k)$; (2) branch node labeling costs $\mathcal{O}(d)$. First we note that, thanks to the hierarchical labeling approach of I$^2$-GNN, all subgraph sampling methods can be applied to address the growing subgraph size problem, such as [5]. Here we instead focus on alleviating the cost from branch node labeling. A simple but efficient idea is to sample a small portion of branch nodes to label. We provide one of the possible implementations: for each root node, we randomly sample fixed-size branch nodes. This implementation makes the branch node labeling complexity a constant factor instead of scaling up with node degree $d$. We evaluate the effectiveness of branch node sampling on datasets with higher average node degree than small molecules, including ENZYMES ($d=3.81$) and IMDB-BINARY ($d=9.77$). We uniformly sample and fix two branch nodes for each root node before training, and disable sampling during test. Please kindly see the result in Table 1 in our response ``Main revisions to the paper" on top. It shows that I$^2$-GNNs with branch node sampling (here we randomly choose two branch nodes) still have a better performance than baselines, and a competitive performance to original I$^2$-GNNs. We add this discussion and the results to Appendix K in the revised paper.
>
>
> [1] Loukas, Andreas. "How hard is to distinguish graphs with graph neural networks?." Advances in neural information processing systems 33 (2020): 3465-3476.
>
> [2] Loukas, Andreas. "What graph neural networks cannot learn: depth vs width." arXiv preprint arXiv:1907.03199 (2019).
>
> [3] Dwivedi, Vijay Prakash, et al. "Graph neural networks with learnable structural and positional representations." arXiv preprint arXiv:2110.07875 (2021).
>
> [4] Wijesinghe, Asiri, and Qing Wang. "A New Perspective on" How Graph Neural Networks Go Beyond Weisfeiler-Lehman?"." International Conference on Learning Representations. 2021.
>
> [5] Zhao, Lingxiao, et al. "From stars to subgraphs: Uplifting any GNN with local structure awareness." arXiv preprint arXiv:2110.03753 (2021).

---

> ### Author Response · Authors · 2022-11-29
> **Would you mind letting us know if our response has addressed your concerns?**
>
> Dear reviewer  Mx4a,
>
> We would like to thank you again for the valuable feedback! Given that there are two weeks left for discussion stage, we would really appreciate it if you could confirm whether our
> response has addressed your concerns.
>
> To brieftly summerize our response, we:
>
> 1. added discussion on the suggested related works, including [1, 2, 3, 4], and explained the difference between them and I$^2$-GNN.
>
> 2. discussed and implemented a branch node sampling method to alleviates the high node degree problem. The performance and runtime can be found at Table 1, 2 of "Main revisions to the paper" at top.
>
> We made a great effort in writing the author response. If you feel there is still any other issue, please kindly let us know and we are happy to follow up with you before the end of the discussion stage.
>
>
> Thanks,
>
> Authors

---

### Official Review · Reviewer_GzCp · 2022-10-25

**Confidence:** 4
**Correctness:** 4
**Technical Novelty And Significance:** 3
**Empirical Novelty And Significance:** 2
**Recommendation:** 6

**Clarity, Quality, Novelty And Reproducibility:**

The paper is well written. If possible, I would recommend to move the discussion about the complexities of the various architecture in the main part of the paper.
The code is provided but I did not run it.

**Strength And Weaknesses:**

The theoretical analysis made in this paper is rather convincing.
The main weakness is the special focus on counting (short) cycles and paths. The expressive power of MPNN can be improved by adding extra-features like it is done in this paper but it is not clear that focusing on cycles and paths is actually the best use of resources. Indeed, the experimental results on real datasets are quite disappointing in this respect as CIN seems to outperform the I2-GNN on ZINC and ogbg-molhiv.

**Summary Of The Paper:**

This paper proposes a new Graph Neural Network architecture enhancing the standard message passing architectures (MPNN). The main idea is to run various MPNNs on rooted subgraphs of the original graph. The authors show that their architecture I2-GNN is more powerful than previous MPNN. They show theoretical results where I2-GNN is able to count longer cycles (3,4, 5 or 6-cycles) and paths (3 or 4-paths). They also show some theoretical limits of I2-GNN. In their last section, they conduct experiments to substantiate their claims.

**Summary Of The Review:**

The paper presents interesting theoretical results but experimental results are weak.

---

> ### Author Response · Authors · 2022-11-17
> **Author response**
>
> We thank the reviewer for their valuable review. In response to the reviewer's concerns:
>
> > **Q1:** ``The main weakness is the special focus on counting (short) cycles and paths. The expressive power of MPNN can be improved by adding extra-features like it is done in this paper but it is not clear that focusing on cycles and paths is actually the best use of resources."
>
> **A1:** As stated in our paper, one motivation of focusing on short cycles is that cycles play an important role in organic chemistry, yet are hardly recognizable by existing GNNs. Different types of rings can greatly impact compounds’ stability, aromaticity and other chemical properties. Another motivation is to draw GNN expressive power community's focus from studying distinguishing power to studying function approximation power, the latter of which usually has more practical value. For example, the simplest 1-WL can already distinguish almost all graphs [1], yet a 1-WL-GNN cannot even count 3-cycles. Studying GNNs' function approximation power for specific targets can help us better understand their expressive power for practical tasks, and paths/cycles can be a starting point towards this direction.
>
>
> > **Q2:**``If possible, I would recommend to move the discussion about the complexities of the various architecture in the main part of the paper."
>
> **A2:**  We thank the reviewer for the suggestion. Due to the current page limitation, we will add the discussion into the main paper if our paper is accepted and it allows additional pages in camera-ready version.
>
> [1] Canonical labelling of graphs in linear average time. SFCS 1979.

---

### Official Review · Reviewer_YNBY · 2022-10-28

**Confidence:** 3
**Correctness:** 3
**Technical Novelty And Significance:** 3
**Empirical Novelty And Significance:** 3
**Recommendation:** 6

**Clarity, Quality, Novelty And Reproducibility:**

The algorithm is novel, and the proof looks correct, although I couldn't check all the details.

**Strength And Weaknesses:**

Strengths

- The proposed algorithm $I^2$-GNN theoretically guarantees the power of counting 6-cycles. This is the first algorithm that can guarantee such power.

- The algorithm's empirical performances support the theoretical results.


Weakness

- The computational complexity of $I^2$-GNN is proportional to the degree of the graph. So, when the degree follows a power law (a heavy tail distribution), the complexity could be much higher than before.

- The empirical study considers QM9 and ZINC. This could be a limited set. It would nice to consider node OGB data set and classification tasks with cora, citeceer, pubmed.

**Summary Of The Paper:**

Message Passing Neural Network (MPNN) is a simple yet efficient class of Graph Neural Networks (GNNs), which have been widely adopted in many previous works. However, it is also known that MPNN has limited representational power due to its simple structure. This paper, in particular, considers subgraph methods. The authors first show that Subgraph MPNNs can count 3-cycles and 4-cycles, but cannot count 5-cycle or any longer cycles at the node level. The cycle counting is important to understand structure since cycles are the basic elements constructing some important substructures. Then, they propose I^2-GNN that can count cycles with lengths less than 7. Based on this, I^2-GNN has a stronger discriminative power than subgraph MPNNs. I^2-GNN outperforms baselines in molecular prediction benchmarks.

**Summary Of The Review:**

This paper has a novel method and its theoretical motivation. However, the experiment section is weak.

---

> ### Author Response · Authors · 2022-11-17
> **Author response**
>
> We thank the reviewer for their insightful review. The followings are responses to the reviewer's comments:
>
> > **Q1:** ``The computational complexity of I$^2$-GNN is proportional to the degree of the graph. So, when the degree follows a power law (a heavy tail distribution), the complexity could be much higher than before."
>
> **A1:** Indeed, unlike the bounded node degree in small molecules, a heavy tail distribution of node degree could cause unexpected high computational cost in some datasets. To alleviate the problem, we discuss and implement a simple but effective branch node sampling strategy.  Please kindly see results in our response ``Main revisions to the paper" on top.  It shows that I$^2$-GNNs with branch node sampling (here we randomly choose and fix two branch nodes) still have a better performance than baselines, and a competitive performance to original I$^2$-GNNs. More importantly, the computational cost is significantly reduced.
>
> > **Q2:** ``The empirical study considers QM9 and ZINC. This could be a limited set. It would nice to consider node OGB data set and classification tasks with cora, citeceer, pubmed."
>
> **A2:** Thank you for the suggestion. We already included the ogbg-molhiv results in Table 5 of the paper. We further test the performance of I$^2$-GNN on datasets from TUD benchmark, as shown in Table 1 in our response ``Main revisions to the paper" on top. The result shows that I$^2$-GNNs also can bring uniform improvements to base GNN and Nested GNN on these high-node-degree datasets.

---

### Official Review · Reviewer_FzhU · 2022-10-30

**Confidence:** 3
**Correctness:** 4
**Technical Novelty And Significance:** 4
**Empirical Novelty And Significance:** 4
**Recommendation:** 8

**Clarity, Quality, Novelty And Reproducibility:**

This paper clearly clarifyies the novelty of the proposed method. The quality of this work is solid and strong in the aspects of both theory and experiments. Source codes and detailed experiment settings in the appendix are provided to ensure good reproducibility.

**Strength And Weaknesses:**

Strengths:
(+) This work makes very solid and novel contributions to developing more expressive graph neural network models. The idea of improving the subgraph counting ability by assigning different identifiers to the root node is interesting and elegant.
(+) Comprehensive theoretical analysis and empirical studies are presented to show the effectiveness of the proposed I^2-GNN model.
(+) The writing of this paper is very clear and well-organized.

Weaknesses:
(-) Some experimental settings are unclear to me in graph substructure counting. For experiments in section 6.2, the prediction targets are cycle or graphlet counts, which are discrete numbers. Do GNN models output continuous numbers to approximate discrete targets and the MAE is evaluated between them?

**Summary Of The Paper:**

This work studies the counting ability of subgraph MPNN models, and present I^2-GNN to increase the counting ability of subgraph MPNNs. The proposed I^2-GNN is able to discriminate 6-cycles in theory and achieves good performance in molecular property prediction.

**Summary Of The Review:**

Overall, this work proposes a novel method in enhancing the representation ability and counting power and of graph neural networks. Both the theorical and experimental contributions of this work are strong. I vote for accepting this work.

---

> ### Author Response · Authors · 2022-11-17
> **Author response**
>
> We thank the reviewer for their thoughtful review as well as for acknowledging our theoretical contribution.
>
> > `` Some experimental settings are unclear to me in graph substructure counting. For experiments in section 6.2, the prediction targets are cycle or graphlet counts, which are discrete numbers. Do GNN models output continuous numbers to approximate discrete targets and the MAE is evaluated between them?"
>
> Yes. The models output continuous predictions to approximate discrete number of substructures, and the MAE is measured between them. This is the same as [1]. We will make it more clear in the revised version.
>
> [1] Chen, Zhengdao, et al. "Can graph neural networks count substructures?." Advances in neural information processing systems 33 (2020): 10383-10395.

---

> > ### Comment · Reviewer_FzhU · 2022-11-19
> > **Thanks for your responses**
> >
> > I appreciate authors' hard work in rebuttal. My question has been addressed and I will keep my original rating of accept.

---

### Public Comment · ~Yuxin_Dong2 · 2022-11-11
**Questions about the claimed linear complexity**

Dear authors,

Thanks for your efforts in presenting this interesting work. However, I have the following questions that confuse me a lot and that I found the reviewers have not yet asked. So I think I should post them here as it may be helpful to the community:
- The authors claimed that "to the best of our knowledge, it is the first linear-time GNN model that can count 6-cycles with theoretical guarantees". However, such a statement is definitely wrong. The proposed subgraph GNN has to enumerate all pair of adjacent nodes and perform message-passing for each pair (see Equation 6), which results in **a complexity equal to $m$ MPNNs** where $m$ is the number of edges in a graph. Therefore, the computational cost should be $O(m(n+m))$, namely $O(m^2)$ for a graph with $n$ nodes and $m$ edges. In other words, the computational complexity scales in proportional to the **square** of the number of edges or the **quartic** of the number of nodes. This cost can even be larger than 3-WL. Even in the bounded degree setting, the complexity is $O(n^2)$ which is *still not linear*. Moreover, the memory cost scales like $O(mn)$. Given this, I am quite confused by the following sentence in the introduction: "I$^2$-GNNs have linear space and time complexity w.r.t. the number of nodes, making it very scalable in real-world applications." I think the contribution of this paper is largely **overclaimed** and even **incorrect**.

- The authors said that prior subgraph-based GNNs can only count cycles with length no more than 4 while I$^2$-GNNs can count cycles with length no more than 6. However, such a statement is **misleading** and may not be true. In particular, their counterexamples (Appendix D.3) for showing that subgraph GNNs cannot count 5-cycles only hold for very restrictive models, such as Cotta et al., 2021. In contrast, most subgraph GNNs can indeed distinguish the two graphs in Appendix D.3, including the following works:
  - Nested GNNs (Zhang et al., 2021). NGNN can distinguish the two graphs as long as the number of outer GNN layers is more than one.
  - ESAN (Bevilacqua et al., 2022). ESAN can distinguish the two graphs using DSS-WL.
  - SUN (Frasca et al., 2022). SUN can distinguish the two graphs since it is more powerful than ESAN.
  - Ordered Subgraph GNN (Qian et al., 2022).

  Given this, it is not clear whether subgraph-based GNNs cannot count 5-cycles or 6-cycles. Note that I$^2$-GNNs requires $O(mn)$ memory and $O(m^2)$ computational cost, which are strictly more costly than all prior subgraph GNNs. If prior works already suffice to count 5-cycles and 6-cycles, there may be of little significance to propose a new architecture with more costs.

---

> ### Author Response · Authors · 2022-11-12
> **Author Response**
>
> Dear Yuxin Dong,
>
> Thank you for your interest in our work and for posting these comments.
>
> (1) It is true that I$^2$-GNNs have to perform $m$ times MPNNs. However, the MPNNs are performed in pre-chosen **subgraphs** rather than the original full graph. For example, one typical choice is
> to choose $k$-hop ego-networks, whose size is about $d^k$ ($d$ is node degree). If the node degree is bounded, the complexity of I$^2$-GNNs should be $O(m d^{k+1})$ instead of $O(m^2)$. It only brings a factor $d$ compared to Subgraph MPNNs. Indeed, in the limiting case of taking a super large $k$ you can make each subgraph the whole graph, then your complexity analysis is correct. However, one major motivation of I$^2$-GNNs is to count 6-cycles (which are important substructures in chemistry), and we only need at most $k=3$ hops to include all $\le 6$-cycles around nodes. Therefore, the subgraph sizes in I$^2$-GNNs are bounded and all our theorems hold in this bounded case. Your statement that our paper is "overclaimed" and "incorrect" is **groundless**.
>
> (2) Our claim (Theorem 3.1) is that **Subgraph MPNNs** cannot count cycles with length no more than 4 at node level. Here Subgraph MPNN is formally defined in Definition 3.1 and is a **subset** of Subgraph GNNs.
> It contains models such as ID-GNNs [1], Nested GNNs [2] without outer GNNs and
> $(N-1)$-reconstruction GNNs [3]. In fact, Subgraph MPNNs can generally represent node-based Subgraph GNNs that perform MPNNs within subgraphs independently (no subgraph-subgraph interactions), and Theorem 3.1 apply to all these kinds of Subgraph GNNs. The examples you provide are Subgraph GNNs using different types of equivariant subgraph-subgraph interactions (Nested GNNs with outer GNNs, ESAN and SUN) as well as using beyond-node-based selection policy (OSAN). Therefore they do not belong to Subgraph MPNNs and are out of the scope of Theorem 3.1. For general Subgraph GNNs
> with arbitrary equivariant operators, it is indeed not known which of them can count 5-cycles.
>
> Further, you may confuse the concepts of **node-level** and **graph-level** counting, which are formally defined in Definitions 2.1 and 2.2. All our theorems are established under node level (which is actually stronger than graph-level counting). Thus, simply **distinguishing the two graphs** in Appendix D.3 is not enough, as it provides no information of cycle counts on particular nodes, and even provides little information on the cycle count of the entire graph. This is why we argue the importance of studying the **function approximation** ability in addition to distinguishing power in GNN expressiveness research. Therefore, although your listed Subgraph GNNs "can indeed distinguish the two graphs in Appendix D.3", they are **not related** to our theorems and claims.
>
> [1] You et al. Identity-aware Graph Neural Networks.
>
> [2] Zhang et al. Nested Graph Neural Networks.
>
> [3] Cotta et al. Reconstruction for Powerful Graph Representations.

---

> > ### Public Comment · ~Yuxin_Dong2 · 2022-11-15
> > **Thank you for the prompt response. Several concerns remain.**
> >
> > Thank you for the prompt and detailed response. First, I would like to acknowledge that the authors have partially addressed my first concern. In particular, I understand what the linear complexity refers to. Concretely, the formal statement would be:
> > > Using $k$-ego-network policy with **a constant** $k$ and assume the graph has **bounded degree** $d$, then the complexity of I$^2$-GNN is bounded by $O(md^{k+1})$ which is linear in $n$.
> >
> > However, I would suggest the authors to revise the abstract and the introduction part since it does not adequately mention the two crucial assumptions, which may confuse readers a lot. This is also why I think the current paper is somewhat overclaimed. In the general case when $d$ and $k$ are not constants, the complexity of I$^2$-GNNs can reach $\Omega(n^4)$ which is even higher than 3-WL. On the other hand, if $k$ is bounded, I$^2$-GNN is **not** strictly powerful than 1-WL (as stated in the third item of the "Main contribution" part). Overall, the property that I$^2$-GNN has linear complexity and the theoretical results that I$^2$-GNN is strictly stronger than MPNNs and partially stronger than 3-WL are under **completely *different* assumptions**.
> >
> > Regarding the second question, I am currently **not satisfied** with the answer.
> > - First, I am confused on why prior works like NGNN, GNN-AK, ESAN, SUN, and OSAN are not subgraph MPNNs, as the aggregation is also based on the message passing.
> > - Second, I do understand the concepts of node-level and graph-level counting and know that node-level counting power implies graph-level one. I would like to emphasize that NGNN, ESAN, SUN, and OSAN can distinguish your counterexample in Appendix D.3 not only in the graph level **but also in the node-level**. It is straightforward to see that the representations of the central nodes are different between the two graphs. In fact, for the graph level cycle counting power, the vanilla subgraph GNN like Cotta et al. (2021) can already distinguish the two graphs in Appendix D.3.
> >
> > In particular, most of the related works on subgraph GNNs (e.g., NGNN, GNN-AK, ESAN, SUN, OSAN) do not fit your definition of subgraph MPNNs. I think the current statement is misleading because it just ignores these related works and does not discuss their counting power. Therefore, it is not known whether I$^2$-GNN is indeed better than prior works and is the first work to count 5/6 cycles efficiently.

---

> > > ### Author Response · Authors · 2022-11-15
> > > **Author response (2.1)**
> > >
> > > Dear Yuxin Dong,
> > >
> > > > "Overall, the property that I$^2$-GNN has linear complexity and the theoretical results that I$^2$-GNN is strictly stronger than
> > > MPNNs and partially stronger than 3-WL are under completely different assumptions."
> > >
> > > Firstly, let us clarify the following: we have claimed that I$^2$-GNN can **count 6-cycles with linear complexity**, and we have claimed that I$^2$-GNN are more powerful than 1-WL and partially stronger than 3-WL in discriminative power. But we have **never** claimed that I$^2$-GNN can **surpass 1-WL/3-WL with linear complexity**. To convince you, we list all sentences in our paper involving “linear complexity”:
> > >
> > > Abstract: *“More importantly, I$^2$-GNNs are proven capable of counting all 3, 4, 5 and 6-cycles, covering common substructures like benzene rings in organic chemistry, while still keeping linear complexity. To the best of our knowledge, it is the first linear-time GNN model that can count 6-cycles with theoretical guarantees.”*
> > >
> > > Introduction: *“Given bounded node degree, I$^2$-GNNs have linear space and time complexity w.r.t. the number of nodes, making it very scalable in real-world applications. To our best knowledge, I$^2$-GNN is the first linear-time GNN model that can count 6-cycles with rigorous theoretical guarantees.”*
> > >
> > > Conclusion: *“The resulting model named I$^2$-GNNs turns out to be able to count at least 6-cycles in linear time with theoretical guarantee.”*
> > >
> > > In all the places, we **clearly state** that I$^2$-GNN can **count 6-cycles with linear complexity**. We again emphasize that the main concentration of our paper is function approximation power (cycle counting function in particular), not graph isomorphism test (discriminative power) as in other expressiveness papers. Our main theorems on counting power are given in Theorems 4.1, 4.2 and 4.3. Proposition 4.1 is just a straightforward side conclusion of I$^2$-GNN’s discriminative power, where the linear complexity obviously does not apply. In fact, any Subgraph GNN cannot surpass 1-WL’s discriminative power in linear complexity, because the subgraph depth must be the whole graph to mimic MPNN’s receptive field for each node.
> > >
> > > Given that no reviewer has given similar concern on our claim of linear complexity and we are sure there is no misleading sentences in our paper, we do not think our claim of linear complexity will confuse the general readers. But we are willing to add one sentence near Proposition 4.1 to emphasize that the linear complexity does not apply there.

---

> > > > ### Public Comment · ~Yuxin_Dong2 · 2022-11-16
> > > > **Further response**
> > > >
> > > > Thank you for your prompt response.
> > > >
> > > > - You said that
> > > > > you must give some restrictions to Subgraph GNNs to do a counting power analysis, otherwise Subgraph GNNs can be too general because their ultimate form is 3-IGN [1], which has exponential complexity.
> > > >
> > > >   That is **not true**. The complexity of 3-IGN is bounded by $O(n^3)$ per layer, see Maron et al. 2019 [1]. Moreover, all Subgraph GNNs I mentioned (e.g., NGNN, GNN-AK, ESAN, SUN, and OSAN) have a complexity of $O(nm)$ for general graphs, or have a linear complexity under the same assumptions of bounded degree plus constant $k$-ego net as your paper.
> > > >
> > > > - In the abstract:  you said
> > > > > I$^2$-GNNs' discriminative power is shown to be strictly stronger than Subgraph MPNNs and partially stronger than the 3-WL test. More importantly, I-GNNs are proven capable of counting all 3, 4, 5 and 6-cycles, covering common substructures like benzene rings in organic chemistry, while still keeping linear complexity.
> > > >
> > > >   I believe these few sentences give readers the impression that I$^2$-GNN is very strong while simultaneously having a linear complexity, which is not true. More importantly, the word *bounded degree* first appears in the "Main contribution" part and the assumption of $k$-ego network with constant $k$ **never appears** in the abstract or introduction. My point is that the linear complexity is emphasized several times in the abstract and introduction, but these crucial assumptions are never presented and seem to be deliberately hidden, which is confusing. Nevertheless, I think this can be easily fixed by presenting the assumptions clearly in the abstract and introduction.
> > > >
> > > > Also, it is nice to see
> > > >
> > > > > we will add emphasis to the paper on the fact that there exists Subgraph GNNs such as GNNAK, DSS-GNN, SUN, OSAN, which do not belong to Subgraph MPNNs. Studying the counting power of more general Subgraph GNNs is another interesting topic which is left for future works.
> > > >
> > > > [1] On the universality of invariant networks. Maron et al. ICML 2019.

---

> > > ### Author Response · Authors · 2022-11-15
> > > **Author response (2.2)**
> > >
> > > Dear Yuxin Dong,
> > >
> > > > “I am confused on why prior works like NGNN, GNN-AK, ESAN, SUN, and OSAN are not subgraph MPNNs, as the aggregation is also based on the message passing.”
> > >
> > > As we said, Subgraph MPNNs are defined by using MPNNs **independently** on subgraphs (please check Definition 3.1 and Equations 3,4,5), where the subgraphs do not exchange information. The key is not only about message passing, but also lies in that the message passing can only happen within each subgraph. For example, DS-GNN (ESAN without information sharing between subgraphs) is Subgraph MPNNs, but DSS-GNN (ESAN that allows information sharing) is not. NGNN without outer layer is Subgraph MPNNs, but NGNN with outer layer is not. GNNAK with one outer GNNAK layer and arbitrary inner MPNN layers is Subgraph MPNNs, but GNNAK with more than one outer GNNAK layers is not. SUN uses all equivariant linear operators among subgraphs and thus is not Subgraph MPNN. OSAN even pools node representation along the subgraph channel, making it essentially different from the models above (which pools within the subgraph).
> > >
> > >
> > > > “I think the current statement is misleading because it just ignores these related works and does not discuss their counting power.”
> > >
> > > We aim to study the counting power of Subgraph MPNNs, **not all possible Subgraph GNNs**. This is stated very clearly in our paper. Not to mention that Subgraph MPNNs already cover a range of popular and pioneering Subgraph GNNs including ID-GNN, NGNN (the original paper does not use outer GNN), DS-GNN and (N-1)-Reconstruction GNN.
> > >
> > > Moreover, we have used almost one entire page (Page 3 to 4) to define and distinguish Subgraph MPNNs, and all our claims are clearly stated and proved using the definition of Subgraph MPNNs. Therefore, we do not understand what makes the current statement misleading. In fact, you must give some restrictions to Subgraph GNNs to do a counting power analysis, otherwise Subgraph GNNs can be too general because their ultimate form is 3-IGN [1], which has exponential complexity.
> > >
> > > > “Therefore, it is not known whether I$^2$-GNN is indeed better than prior works and is the first work to count 5/6 cycles efficiently.”
> > >
> > > We didn’t claim I$^2$-GNN is better than all prior works. What we proved is that I$^2$-GNN is more powerful than Subgraph MPNNs in counting power. Besides, what we claimed in the abstract as well as the main paper is that I$^2$-GNN is the first linear-time GNN model that can **count 6-cycles with theoretical guarantees**. There may exist other Subgraph GNNs with 6-cycle counting power, but proving it is nontrivial, and is out of the scope of our paper.
> > >
> > > Nevertheless, given the discussion, we will add emphasis to the paper on the fact that there exists Subgraph GNNs such as GNNAK, DSS-GNN, SUN, OSAN, which do not belong to Subgraph MPNNs. Studying the counting power of more general Subgraph GNNs is another interesting topic which is left for future works.
> > >
> > > [1] Frasca, Fabrizio, et al. "Understanding and extending subgraph gnns by rethinking their symmetries.”

---

### Author Response · Authors · 2022-11-17
**Main revisions to the paper**

We thank all the reviewers for their constructive comments. In response to reviewers' comments and concerns, we make some revisions to the paper, which are highlighted in blue in the main text. In the following we list the main changes in the revised paper.

- **More discussions on related works.**  We add discussions on [1, 2, 3, 4, 5, 6] in the Related Works section.

- **Study of sampling method for high node-degree graphs.** Some of the reviewers suggested to discuss how I$^2$-GNNs can be applied to graphs with high average node degree. Here we study a simple but effective sampling method to reduce the training time: randomly sample fixed-size branch nodes for each root node. It can reduce the complexity of labeling branch nodes to a controlled constant factor. We further validate the effectiveness of branch node sampling on datasets with higher node degree than small molecules, including ENZYMES ($d=3.81$) and IMDB-BINARY ($d=9.77$). We define the model I$^2$-sample-GNN to be an I$^2$-GNN that randomly samples and fixes two branch nodes for each root node before training. The sampling is disabled when testing. We choose subgraph depth to be $3$ for ENZYMES and $2$ for IMDB-BINARY. The 10-fold cross validation results are shown in Table 1 below. We can see that I$^2$-sample-GNNs still attain a better performance than baselines, and have a competitive performance to original I$^2$-GNNs. Moreover, we compare the number of parameters and the training/test time for these models in Table 2. As we expected, I$^2$-sample-GNNs can significantly reduce the training time, especially for high-node-degree dataset IMDB-BINARY. The test time remains similar because we do not sample during test (the mean pooling over branches resolves the size generalization). Detailed discussion on branch node sampling and experiments can be found in Appendix K of the revised paper.


Table 1: Test accuracy (\%) $\pm$ one standard deviation on TUD benchmark. I$^2$-sample-GNN refers to I$^2$-GNNs that randomly choose and fix two branch nodes to label before training.
|  Dataset   | Base GNN  | Nested GNN | I$^2$-GNN | I$^2$-sample-GNN |
|  ----  | :----:  |  :----:  | :----:  | :----: |
| ENZYMES | 27.3 $\pm$ 7.8 | 31.7 $\pm$ 3.7 | 35.8 $\pm$ 7.1 | 33.8 $\pm$ 5.9 |
| IMDB-BINARY | 70.2 $\pm$ 5.1 | 71.4 $\pm$ 5.9 | 73.5 $\pm$ 3.0 | 72.7 $\pm$ 3.9 |

Table 2: \#parameters/training time per 200 epochs (s)/inference time per epoch (ms) on TUD benchmark. Training/inference time are estimated on full dataset.
|  Dataset   | Base GNN  | Nested GNN | I$^2$-GNN | I$^2$-sample-GNN |
|  ----  | :----:  |  :----:  | :----:  | :----: |
| ENZYMES | 24k/19.45/59.09 | 25k/30.31/107.44 | 34k/62.82/213.44 | 34k/45.68/213.03 |
| IMDB-BINARY |  24k/31.19/88.49 | 25k/65.28/238.91 | 34k/355.98/1273.30 | 34k/**100.33**/1270.62 |



[1] Loukas, Andreas. "How hard is to distinguish graphs with graph neural networks?." Advances in neural information processing systems 33 (2020): 3465-3476.

[2] Loukas, Andreas. "What graph neural networks cannot learn: depth vs width." arXiv preprint arXiv:1907.03199 (2019).

[3] Dwivedi, Vijay Prakash, et al. "Graph neural networks with learnable structural and positional representations." arXiv preprint arXiv:2110.07875 (2021).

[4] Wijesinghe, Asiri, and Qing Wang. "A New Perspective on" How Graph Neural Networks Go Beyond Weisfeiler-Lehman?"." International Conference on Learning Representations. 2021.

[5] Zhang, Muhan, et al. "Labeling trick: A theory of using graph neural networks for multi-node representation learning." Advances in Neural Information Processing Systems 34 (2021): 9061-9073.

[6] Zhang, Muhan, and Yixin Chen. "Link prediction based on graph neural networks." Advances in neural information processing systems 31 (2018).

---

### Decision · Program_Chairs · 2023-01-20

**Decision:**

Accept: poster

**Justification For Why Not Higher Score:**

Modest novelty.

**Justification For Why Not Lower Score:**

The reviewers raised concerns including comparison of the idea of using multiple identifiers for node pairs (of edges) with labelling tricks, justification of novelty given other expressive GNNs, computational complexity and scalability to denser graphs, the special focus on counting (short) cycles and paths, need for additional datasets. The authors did a good job addressing reviewers’ comments and adding suggested experiments, resulting in overall positive assessment of reviewers after the response period.

**Metareview: Summary, Strengths And Weaknesses:**

This work studies the counting ability of subgraph MPNN models, and present I^2-GNN to increase the counting ability of subgraph MPNNs. The main idea is to run various MPNNs on rooted subgraphs of the original graph. For each rooted subgraph centered at node i, they construct d(i) subgraphs, where d(i) is the degree of the center node, defining a mapping between the generated subgraphs and the neighbors of the root node. Then, they assign a unique identifier to the center node i, and another unique identifier to one of the neighbors of the center node, for each generated subgraph. They use MPNNs in each subgraph to construct the graph's final representation, followed by a readout layer. The proposed I^2-GNN is able to discriminate up to 6-cycles in theory and achieves good performance in molecular property prediction. In contrast, the other Subgraph MPNNs cannot count such long cycles. Based on this, I^2-GNN has a stronger discriminative power than subgraph MPNNs. I^2-GNN has linear space and time complexity provided the node degree is bounded, and outperforms baselines in molecular prediction benchmarks. The reviewers raised concerns including comparison of the idea of using multiple identifiers for node pairs (of edges) with labelling tricks, justification of novelty given other expressive GNNs, computational complexity and scalability to denser graphs, the special focus on counting (short) cycles and paths, need for additional datasets. The authors did a good job addressing reviewers’ comments and adding suggested experiments, resulting in overall positive assessment of reviewers after the response period.

**Note From Pc:**

if the above contains the word "oral" or "spotlight" please see: "oral" presentation means -> notable-top-5% and "spotlight" means -> notable-top-25%. As stated in our emails, we are disassociating presentation type from AC recommendations